# ATR is required to complete meiotic recombination in mice

Sarai Pacheco[1,2], Andros Maldonado-Linares[1,2], Marina Marcet-Ortega[1,2], Cristina Rojas[1,2], Ana Martínez-Marchal[1,2], Judit Fuentes-Lazaro[1,2], Julian Lange[3,4], Maria Jasin[5], Scott Keeney [3,4], Oscar Fernández-Capetillo[6], Montserrat Garcia-Caldés[1,2] & Ignasi Roig [1,2]

Precise execution of recombination during meiosis is essential for forming chromosomally-balanced gametes. Meiotic recombination initiates with the formation and resection of DNA double-strand breaks (DSBs). Cellular responses to meiotic DSBs are critical for efficient repair and quality control, but molecular features of these remain poorly understood, particularly in mammals. Here we report that the DNA damage response protein kinase ATR is crucial for meiotic recombination and completion of meiotic prophase in mice. Using a hypomorphic *Atr* mutation and pharmacological inhibition of ATR in vivo and in cultured spermatocytes, we show that ATR, through its effector kinase CHK1, promotes efficient RAD51 and DMC1 assembly at RPA-coated resected DSB sites and establishment of inter-homolog connections during meiosis. Furthermore, our findings suggest that ATR promotes local accumulation of recombination markers on unsynapsed axes during meiotic prophase to favor homologous chromosome synapsis. These data reveal that ATR plays multiple roles in mammalian meiotic recombination.

[1] Genome Integrity and Instability Group, Institut de Biotecnologia i Biomedicina, Universitat Autònoma de Barcelona, Cerdanyola del Vallès 08193, Spain. [2] Department of Cell Biology, Physiology and Immunology, Universitat Autònoma de Barcelona, Cerdanyola del Vallès 08193, Spain. [3] Molecular Biology Program, Memorial Sloan Kettering Cancer Center, New York, NY 10065, USA. [4] Howard Hughes Medical Institute, Memorial Sloan Kettering Cancer Center, New York, NY 10065, USA. [5] Developmental Biology Program, Memorial Sloan Kettering Cancer Center, New York, NY 10065, USA. [6] Genomic Instability Group, Spanish National Cancer Research Centre, Madrid 28029, Spain. Correspondence and requests for materials should be addressed to I.R. (email: ignasi.roig@uab.cat)

Before the first meiotic division, hundreds of programmed DNA double-strand breaks (DSBs) are formed throughout the genome by the SPO11 protein[1]. The repair of these lesions by homologous recombination promotes homologous chromosome synapsis and crossover formation. Crossovers are essential for proper chromosome segregation and thus for preventing aneuploidy in gametes[2].

The cellular machinery that senses meiotic DSBs and regulates their repair is similar to the surveillance proteins that monitor DNA integrity in somatic cells and that, as a result of DNA damage, activate repair pathways, arrest cell cycle progression, and, if necessary, induce apoptosis[3,4]. A major one of these regulators in mice is the ataxia telangiectasia mutated and Rad-3 related (ATR) protein kinase[3,4]. ATR is activated by the presence of single-stranded DNA (ssDNA), which mainly arises from stalled replication forks or resected DSBs. ssDNA is rapidly covered by the ssDNA-binding protein complex RPA, which recruits ATR and its cofactors. ATR activation promotes stabilization and restart of stalled replications forks, DNA repair, and cell cycle arrest[3].

In mammalian meiosis, ATR accumulates on unsynapsed chromosome axes, such as the heterologous parts of the X and Y chromosomes[5]. This accumulation causes chromatin alterations that, in males, condense the sex chromosomes into a distinct chromatin domain known as the sex body. Such heterochromatinization leads to the transcriptional silencing of these regions, referred to as meiotic sex chromosome inactivation (MSCI)[6]. MSCI is indispensable for the completion of meiotic prophase, as sex-body failure provokes a mid-prophase arrest[7]. Although DSBs may contribute to this mode of ATR action, DSB formation is not strictly required as ATR can promote a similar response to unsynapsed axes in the absence of meiotic DSBs[8,9].

Whether ATR also functions in response to meiotic DSBs has remained obscure. In yeast, the ATR ortholog Mec1 is involved in directing meiotic cell cycle progression, promoting meiotic recombination, and controlling crossover formation[10]. In plants, ATR also promotes meiotic recombination by regulating recombinase DMC1 deposition at resected DSBs[11]. In mouse spermatocytes, ATR and its cofactors co-localize with other DNA repair proteins at sites undergoing homologous recombination, suggesting that ATR functions in meiotic recombination[12]. However, this hypothesis has not been formally tested, in part because abolition of Atr expression is embryonically lethal[13]. We overcame this challenge in this study by diminishing ATR function using genetic and pharmacological tools.

Our findings revealed previously invisible functions for ATR in proper loading of strand-exchange proteins at DSBs, correct timing of crossover formation, elongation of the synaptonemal complex (SC, the zipper-like proteinaceous structure that juxtaposes homologous chromosomes), and proper accumulation of recombination markers on the axes of unsynapsed chromosomes.

## Results

**Using the Seckel mouse model to study meiotic ATR functions.** We used a previously described hypomorphic mutation that reduces ATR expression in most tissues[14]. This allele mimics a mutation found in some human patients with Seckel syndrome (OMIM 210600), which is characterized by dwarfism, microcephaly, mental retardation, and a beak-like nose[15]. The mutation is a synonymous A to G transition in exon 9 that causes frequent skipping of this exon during splicing, thereby severely reducing Atr expression[14,16]. In adult male mice homozygous for this "Seckel" allele (Atr$^S$), whole-testis Atr mRNA – but not protein – levels were substantially reduced, as previously reported[14] (Fig. 1a and Supplementary Fig. 1a). Importantly, we were unable to

detect by immunoblot the ~70 kDa-truncated ATR protein expected from the exon 9-deleted transcript in Seckel mouse testis (Supplementary Fig. 1a). Moreover, even if expressed, the truncated ATR form would be unlikely to compensate for ATR function because it would lack the kinase catalytic domain. Testes from adult Atr$^{S/S}$ mice were much smaller than wild type ($26.0 \pm 11.1\%$ of wild type, at 2–4 months of age, mean ± SD, $N = 20$). This difference may be attributable to the dwarfism of Seckel mice[14] rather than a meiotic arrest. Indeed, the cellular composition of seminiferous tubules appeared grossly normal (Supplementary Fig. 1b) and a previous study found that Atr$^{S/S}$ mice produce functional sperm[14].

Meiotic prophase stages of surface-spread spermatocytes can be determined by the cytological patterns of axis and SC components. At leptonema, SYCP3 protein begins to form the axis of each chromosome (Supplementary Fig. 1c). Synapsis begins at zygonema and leads to the formation of the SC containing SYCP1. At pachynema, autosomal homologous chromosomes are fully synapsed and SYCP3 and SYCP1 proteins completely co-localize. At diplonema, SCs disassemble but homologous chromosomes remain held together by chiasmata. Cytological evaluation of Atr$^{S/S}$ germ cells revealed that both the total percent of cells in meiotic prophase and the fractions of meiotic cells within each prophase stage were similar to wild type (Supplementary Fig. 1d). Taken together, these results show that Atr$^{S/S}$ cells are capable of progressing through meiosis and completing spermatogenesis.

We sought evidence of reduced ATR function in Atr$^{S/S}$ spermatocytes. The only known meiotic role of ATR is in sex-body formation and MSCI at pachynema[5,7]. Since sex-body formation was indistinguishable in wild type and Seckel mice (wild type: $0.4 \pm 0.6\%$, $N = 318$; Atr$^{S/S}$: $2.3 \pm 1.2\%$, $N = 389$; $p = 0.183$, one-way ANOVA), we analyzed the localization pattern of ATR and the intensity of four ATR-dependent sex-body markers in pachytene Atr$^{S/S}$ spermatocytes (Fig. 1b, c and Supplementary Fig. 1e). ATR localizes to the unsynapsed axes of the sex chromosomes, where it phosphorylates the histone H2AX to form γH2AX, and HORMAD2 on S271 (pHORMAD2), among other proteins[5]. With the help of MDC1, ATR spreads to the XY chromatin, where it continues to phosphorylate H2AX[17]. Finally, other sex-body markers such as SUMO-1 are incorporated into the XY chromatin[17]. In wild type, most cells contained ATR localized to both the chromatin and the axes of the sex chromosomes. In contrast, most Atr$^{S/S}$ cells displayed ATR only on the XY axes. The staining pattern in Seckel cells for the ATR-dependent sex-body markers studied (γH2AX, pHORMAD2, MDC1, and SUMO-1, see examples for γH2AX in Fig. 1d, e) was indistinguishable from control cells. However, we observed a significant change in total signal intensity in Atr$^{S/S}$ compared with wild type for each of them. Notably, although most Atr$^{S/S}$ cells lacked ATR in XY chromatin, we detected the other sex-body markers there. Signal intensities of γH2AX and SUMO-1 were reduced, but pHORMAD2 and MDC1 signal were higher, possibly reflecting a compensatory mechanism. Despite these alterations, MSCI appears to be unaffected, as the X-linked gene Zfx was efficiently silenced in pachytene Atr$^{S/S}$ spermatocytes (Supplementary Fig. 1f, g). We conclude that wild-type levels of ATR function are not needed to initiate and/or maintain MSCI. Importantly, however, our findings demonstrate that homozygosity for the Seckel mutation causes mild changes in ATR function in spermatocytes.

To further test this conclusion, we analysed other known ATR target proteins in the sex body. ATR phosphorylates multiple proteins involved in DNA repair, cell cycle control or replication origin firing[3]. Some of these are phosphorylated in the sex body, although it is not clear if they have any MSCI function, like

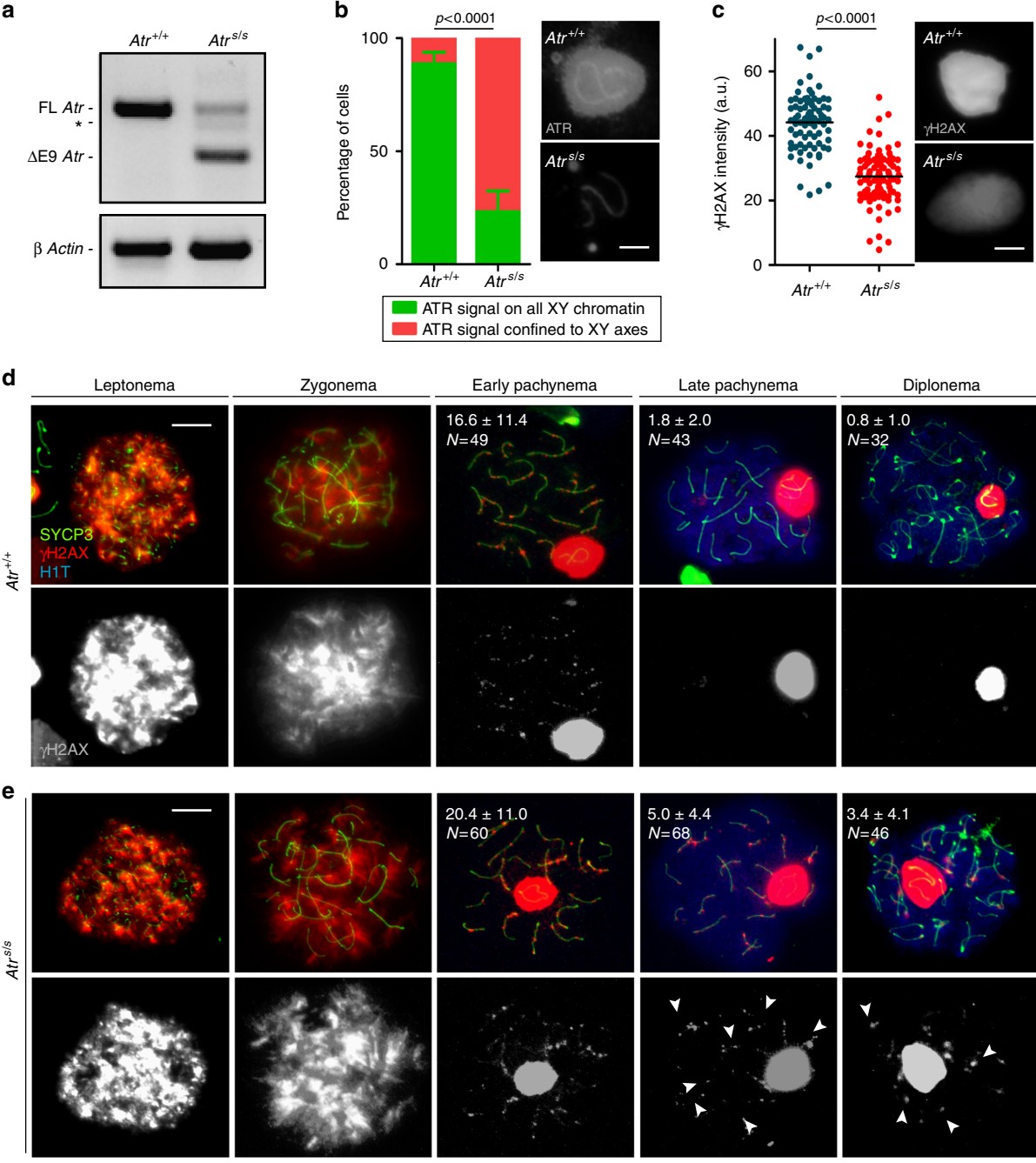

**Fig. 1** Seckel mouse spermatocytes exhibit more γH2AX patches than wild-type cells. **a** RT-PCR using primers that anneal to *Atr* exons 8 and 10. *Atr*$^{S/S}$ testis exhibits two main RT-PCR products, one corresponding to full-length *Atr* (FL *Atr*, 477 bp), which is substantially reduced, and another corresponding to *Atr* lacking exon 9 (ΔE9 *Atr*, 284 bp). Asterisk denotes RT-PCR product resulting from the use of a cryptic splice donor site[16]. RT-PCR for *β Actin* is also provided as a control. **b** Percentage of cells exhibiting ATR staining extended to X and Y chromatin or confined to the X and Y chromosome axes in *Atr*$^{+/+}$ and *Atr*$^{S/S}$ ($N = 284$ and $N = 282$, respectively). Columns and lines indicate the mean and standard deviation (SD) from analysis performed on three wild-type and three mutant mice. Images show ATR straining on the sex body in *Atr*$^{+/+}$ and *Atr*$^{S/S}$ pachytene spermatocytes. Images were captured with the same exposure time. Scale bar = 2 μm. **c** Quantification of the intensity of γH2AX staining on the sex body in arbitrary units (a.u.). Horizontal black lines denote the means. Images show representative sex bodies from *Atr*$^{+/+}$ and *Atr*$^{S/S}$ pachytene spermatocytes immunostained against γH2AX. Images were captured with the same exposure time. Scale bar = 2 μm. Representative images of *Atr*$^{+/+}$ (**d**) and *Atr*$^{S/S}$ (**e**) spermatocytes at different stages of meiotic prophase immunostained against SYCP3, γH2AX, and H1T. Scale bars = 10 μm. Numbers represent the mean ± SD of γH2AX patches found in each stage and genotype. *N* denotes the number of cells analyzed. H1T incorporation marks spermatocytes after mid-pachynema[62]. Arrowheads indicate examples of aberrant autosomal γH2AX patches in *Atr*$^{S/S}$ spermatocytes. The differences between controls and *Atr*$^{S/S}$ for γH2AX patch numbers at late pachynema and diplonema were statistically significant ($p = 0.0001$ and $p = 0.0007$, respectively, $t$ test)

phosphorylation of RPA32 on S33 (pRPA) or phosphorylation of CHK1 on S317 (pCHK1)[18]. Both markers were altered in Seckel mouse cells. Whereas pRPA in wild-type cells was almost exclusively found evenly covering the chromatin of the X and Y chromosomes, ~30% of $Atr^{S/S}$ spermatocytes showed pRPA staining that accumulated more predominantly in the vicinity of the X and Y axes, without covering the whole XY chromatin (Supplementary Fig. 1h). Similarly, about half of $Atr^{S/S}$ spermatocytes had a similar, more axis-restricted staining pattern for pCHK1, unlike the broader chromatin staining seen in wild type (Supplementary Fig. 1h). Moreover, overall pCHK1 intensity was visibly higher in wild-type cells. These phosphorylated forms thus resemble the localization of ATR itself in Seckel mouse cells (Fig. 1b). These findings corroborate that the Seckel mutation impairs ATR function in spermatocytes.

**Modest recombination defects of Seckel mouse spermatocytes.** $Atr^{S/S}$ cells displayed subtle recombination defects. First, we detected persistent autosomal γH2AX patches in pachytene spermatocytes. During early prophase of meiosis, histone H2AX is phosphorylated by ATM in response to meiotic DSBs, which are repaired as prophase progresses[8,19,20]. In control mice, autosomal γH2AX progressively disappeared during prophase (Fig. 1d). From pachynema onward, γH2AX was mostly confined to the sex body, but a few γH2AX patches were visible on the autosomes, presumably at remaining unrepaired DSB sites. In $Atr^{S/S}$ spermatocytes, the numbers of autosomal γH2AX patches at early pachynema, late pachynema, and diplonema were higher than in wild type (Fig. 1e), suggesting delayed repair kinetics. Alternatively, Seckel spermatocytes might have an increased number of late-forming DSBs.

Second, whereas autosomal synapsis was unaffected, synapsis of sex chromosomes was impaired in $Atr^{S/S}$ cells (Supplementary Fig. 1c). Since the XY pair is almost completely heterologous except for the small pseudoautosomal region (PAR), sex-chromosome synapsis must be mediated by DSBs occurring within the PAR[21]. Two-fold more $Atr^{S/S}$ spermatocytes displayed unsynapsed sex chromosomes at pachynema than control cells (12.2% in $Atr^{+/+}$ ($N = 197$) vs. 25.1% in $Atr^{S/S}$ ($N = 235$), $p = 0.0013$, Fisher's exact test). $Atr^{+/S}$ spermatocytes were similar to wild type (14.9% unsynapsed X and Y ($N = 131$), $p = 0.2254$, Fisher's exact test).

To further investigate ATR function in meiotic recombination, we examined RAD51 and DMC1, which associate with resected DSBs and catalyze strand exchange[22,23]. We quantified these markers at progressive stages of meiotic prophase (Fig. 2a, b, Supplementary Table 1). In wild type, RAD51 and DMC1 foci were most abundant at leptonema, then progressively decreased through diplonema, when most foci localized to the non-homologous portions of the sex chromosomes. RAD51 and DMC1 focus numbers were higher in pachtyene $Atr^{S/S}$ spermatocytes than in wild-type controls, consistent with the γH2AX results. In leptonema through zygonema, in contrast, fewer foci were observed in the mutant. Thus, there appear to be temporally distinct defects in recombination in the $Atr^{S/S}$ spermatocytes.

The reduced focus counts in early spermatocytes could reflect either a deficiency in the formation or early processing of DSBs, or defective loading of RAD51 and DMC1. To distinguish between these possibilities, we examined early steps of DSB formation and processing. First, we studied whole-testis DSB levels in $Atr^{S/S}$ mice

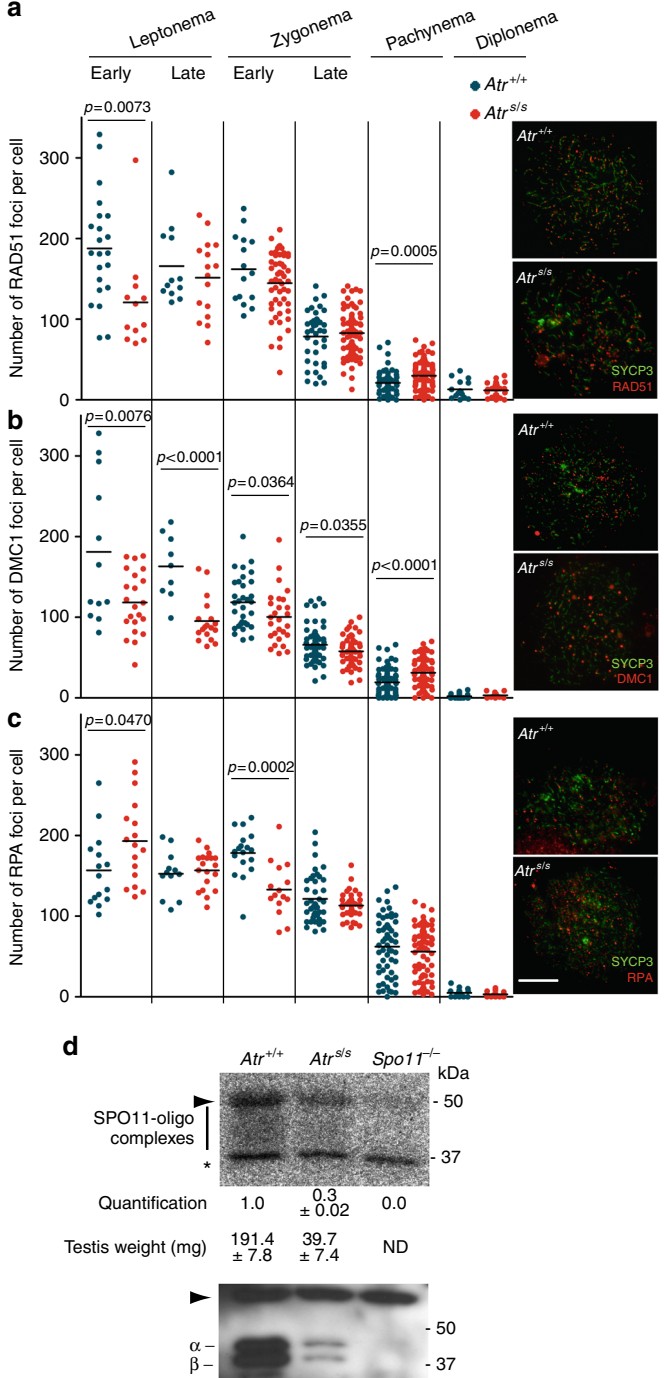

**Fig. 2** Normal DSB formation but altered numbers of recombination markers in $Atr^{S/S}$ cells. **a–c** Left panels, quantification of total foci per spermatocyte for RAD51 (**a**), DMC1 (**b**), or RPA (**c**) at the indicated stages. Each point is the focus count for a single cell. Horizontal lines denote the mean. Right panels, representative images of spermatocytes stained for the indicated proteins. Scale bar = 10 μm. All $p$ values are from $t$ test pairwise comparisons; no $p$ value is stated if the comparison was not statistically significant ($p > 0.05$). **d** Quantification of SPO11-oligonucleotide complexes. A representative experiment is shown. Anti-SPO11 immunoprecipitates from two testes of each genotype were labeled with terminal transferase and [α-$^{32}$P]-dCTP, resolved by SDS-PAGE, and detected by autoradiography (top) and western blotting analysis (bottom). Vertical line indicates the signal from SPO11-oligonucleotide complexes; asterisk indicates non-specific signal from the labeling reaction; positions of the two major SPO11 splicing isoforms, α and β are shown; and arrowheads denote the migration position of immunoglobulin heavy chain. SPO11-oligonucleotide signal and testis weights are indicated (mean ± SD, $N = 2$). Note that $Atr^{S/S}$ testes express both SPO11α and SPO11β, unlike mutants that experience arrested meiotic progression and lack SPO11α as a consequence[37,63,64]

by analyzing SPO11-oligonucleotide complexes, which are a direct readout of SPO11 function[24]. Unlike other mouse mutants with small testes[25–27], $Atr^{S/S}$ mice do not display prophase arrest, so the cellular composition (i.e., fractions of cell types) of their testes is similar to wild type (Supplementary Fig. 1b–d). Thus, to account for the dwarfism of Seckel mice[14], we compared SPO11-oligonucleotide signal in $Atr^{S/S}$ with wild type after normalization by testis weight. By this measure, we did not observe a reduction in SPO11 activity in $Atr^{S/S}$ testes; if anything, testis-weight-normalized SPO11-oligonucleotide levels were slightly elevated in mutant testes (1.6 ± 0.5 fold, mean ± SD, $N = 2$) (Fig. 2d). These findings suggest that SPO11-induced DSB numbers are not reduced in $Atr^{S/S}$ spermatocytes.

Second, we examined the formation of RPA foci, which decorate ssDNA at resected DSB sites prior to the loading of RAD51 and DMC1[22]. In control spermatocytes, RPA focus numbers peaked in zygonema, then progressively decreased (Fig. 2c, Supplementary Table 1). RPA focus counts in $Atr^{S/S}$ spermatocytes were elevated at early leptonema. A straightforward interpretation of the altered RPA, RAD51, and DMC1 focus numbers is that RPA tends to persist longer at DSB sites in $Atr^{S/S}$ cells because of impaired loading of RAD51 and DMC1.

Physical linkages (chiasmata) between homologous chromosomes are crucial for accurate chromosome segregation at metaphase I. At least one crossover per bivalent is required to ensure proper orientation on the first meiotic spindle[28]. Because yeast Mec1 participates in controlling the number and distribution of crossovers[29,30], we investigated crossover formation in Seckel spermatocytes. In mice, several proteins have been involved in crossover formation, among them RNF212, which is the homolog of yeast Zip3. RNF212 has been implicated in designating CO sites by stabilizing key recombination proteins in a subset of recombination nodules[31]. Interestingly, $Atr^{S/S}$ spermatocytes presented more RNF212 foci than control cells on average, and these foci tended to accumulate in clusters along stretches of the SCs (Fig. 3a, Supplementary Table 1).

We also examined MLH1, which is required for the formation of the majority (~90%) of crossovers and which cytologically marks most crossover-designated sites at pachynema[32]. Pachytene $Atr^{S/S}$ spermatocytes displayed one less MLH1 focus on average than controls (Fig. 3b), attributable to a significant increase in the number of autosomal bivalents lacking an MLH1 focus (Fig. 3c). However, we detected no difference in the number of achiasmate autosomal bivalents at diplonema (Fig. 3d). Overall, the RNF212 and MLH1 focus analysis may indicate that MLH1 focus formation is delayed in $Atr^{S/S}$ cells or that ATR regulates the number of crossovers that are repaired by MLH1-dependent and -independent pathways.

Crossover control mechanisms regulate not only the number of crossovers but also their location[33]. The presence of one crossover inhibits the occurrence of another in its vicinity by a mechanism known as interference, which leads to a non-random distribution of crossovers[33,34]. We asked whether ATR reduction affected the distribution of MLH1 foci. We found a similar distribution along autosomal SCs in $Atr^{S/S}$ and wild type (Fig. 3e). In addition, the strength of cytological interference quantified using the gamma distribution[34] was similar in both genotypes (Fig. 3f). Thus, although wild-type ATR levels are required for proper MLH1 focus numbers at pachynema, focus distribution is not substantially altered in $Atr^{S/S}$ cells.

**ATR inhibition in vivo impairs meiotic recombination.** Seckel mice do not completely lose $Atr$ expression (Fig. 1a)[14]. Thus, to further assess the role of ATR in meiotic recombination, we established an experimental system to chemically inhibit ATR.

AZ20 is a selective ATR inhibitor that has been developed as a potential anti-cancer drug[35]. It inhibits ATR function in vitro and in cell culture systems, and its oral administration to mice carrying xenografted adenocarcinoma cells reduces tumor growth[35].

Wild-type adult males were dosed orally with AZ20 or vehicle solution (DMSO) for 3 days, the length of time normally required for a spermatocyte that initiates meiosis to complete homologous chromosome synapsis in vivo[36]. In control mice, 13.3% of testis cells were SYCP3-positive; of these, 81.5% were in pachynema or diplonema (Fig. 4a, b). AZ20-treated mice had fewer SYCP3-positive spermatocytes and altered proportions of prophase stages, including an increased fraction of zygotene cells ($p < 0.0001$, Fisher's exact test) and an almost complete loss of diplotene cells ($p < 0.0001$, Fisher's exact test, see below) (Fig. 4a, b). We analyzed sex-body formation, judged by γH2AX deposition on the X and Y chromatin of pachytene cells, as a readout of ATR function. Compared with controls, AZ20-treated mice had 13-fold more pachytene cells lacking a sex body (DMSO: 0.25 ± 0.35%, $N = 400$; AZ20: 3.25 ± 0.35%, $N = 400$; Mean ± SD, $p = 0.014$, one-way ANOVA, Fig. 4d). These data indicate that in vivo treatment with AZ20 inhibits ATR activity in meiosis and that this inhibition affects meiotic progression.

Next, we analyzed sex-chromosome synapsis and recombination markers. Similar to results with $Atr^{S/S}$ mice, AZ20 treatment caused a more than two-fold increase in the frequency of unsynapsed sex chromosomes at pachynema (Fig. 4c). AZ20-treated spermatocytes exhibited fewer RAD51 foci during leptonema and early zygonema, and, as observed in $Atr^{S/S}$ spermatocytes, the decrease in RAD51 focus number was accompanied by a significant increase in the number of RPA foci at early leptonema (Fig. 4e, Supplementary Table 2). These results suggest that RAD51 association with resected DSBs is delayed in spermatocytes from AZ20-treated males, which could in turn affect sex chromosome synapsis.

To corroborate the ATR role in the dynamics of late recombination foci, we treated adult males with AZ20 or DMSO for 7 days, the time required for a leptotene spermatocyte to reach mid/late-pachynema. This longer treatment resulted in a slightly reduced percentage of SYCP3-positive spermatocytes and an increased number of apoptotic cells per tubule (Fig. 4a, b, and f). The prolonged treatment resulted similarly in a large increase in the fraction of pachytene spermatocytes lacking a sex body (DMSO: 0.5 ± 0.7%, $N = 200$; AZ20-treated: 14.5 ± 0.7%, $N = 200$, Mean ± SD, $p = 0.0002$, one-way ANOVA). Importantly, treated mice lacked diplotene spermatocytes (Fig. 4b), in agreement with a previous report that ATR function is required to exit pachynema[5]. Similar to Seckel mice, AZ20-treated animals accumulated more RAD51 foci at pachynema, corroborating the requirement of ATR for timely repair of DSBs (Fig. 4e, Supplementary Table 2). In addition, AZ20-treated mice accumulated RNF212 at early and late pachynema (Fig. 4g, Supplementary Table 2) and displayed, on average, one less MLH1 focus than controls (Fig. 4h). The lack of diplotene spermatocytes in AZ20-treated mice precluded study of achiasmate bivalents. As in $Atr^{S/S}$ spermatocytes, the distribution of MLH1 foci was not substantially altered in AZ20-treated animals and MLH1 interfocal distances displayed positive interference (Fig. 4i, j). Furthermore, modeling interfocal distances to a gamma distribution gave lower values of cytological interference than the controls, but within the range considered normal in other studies[20,34].

In sum, the results from in vivo treatment of wild-type mice recapitulated our findings with the Seckel mouse model, with some differences that can be attributed to a greater decrease in ATR function from AZ20 treatment. Specifically, unlike $Atr^{S/S}$ mice, AZ20-treated mice displayed meiotic prophase arrest and

more severe alterations to focus counts for recombination markers (Supplementary Tables 1 and 2). Also, whereas almost all pachytene $Atr^{S/S}$ spermatocytes displayed a sex body, 15% of pachytene spermatocytes from mice treated 7 days with AZ20 did not. Thus, reduced ATR activity may prevent sex-body formation, which would cause pachytene arrest and an increase in apoptosis. Moreover, we showed previously that abnormal accumulation of ATR on the XY pair may be sufficient to phosphorylate H2AX but insufficient to implement silencing[37]. Thus, we postulate that the arrest observed after in vivo treatment with AZ20 may be the

consequence of a failure to silence the sex chromosomes at pachynema.

**In vitro ATR inhibition impairs meiotic prophase progression.** Although AZ20 treatment in vivo appeared to attenuate ATR activity more than the Seckel mutation did, the accumulation of γH2AX on the XY chromatin in most treated spermatocytes indicated that ATR function was not fully blocked. This might be attributable to the blood–testis barrier, which controls entry of

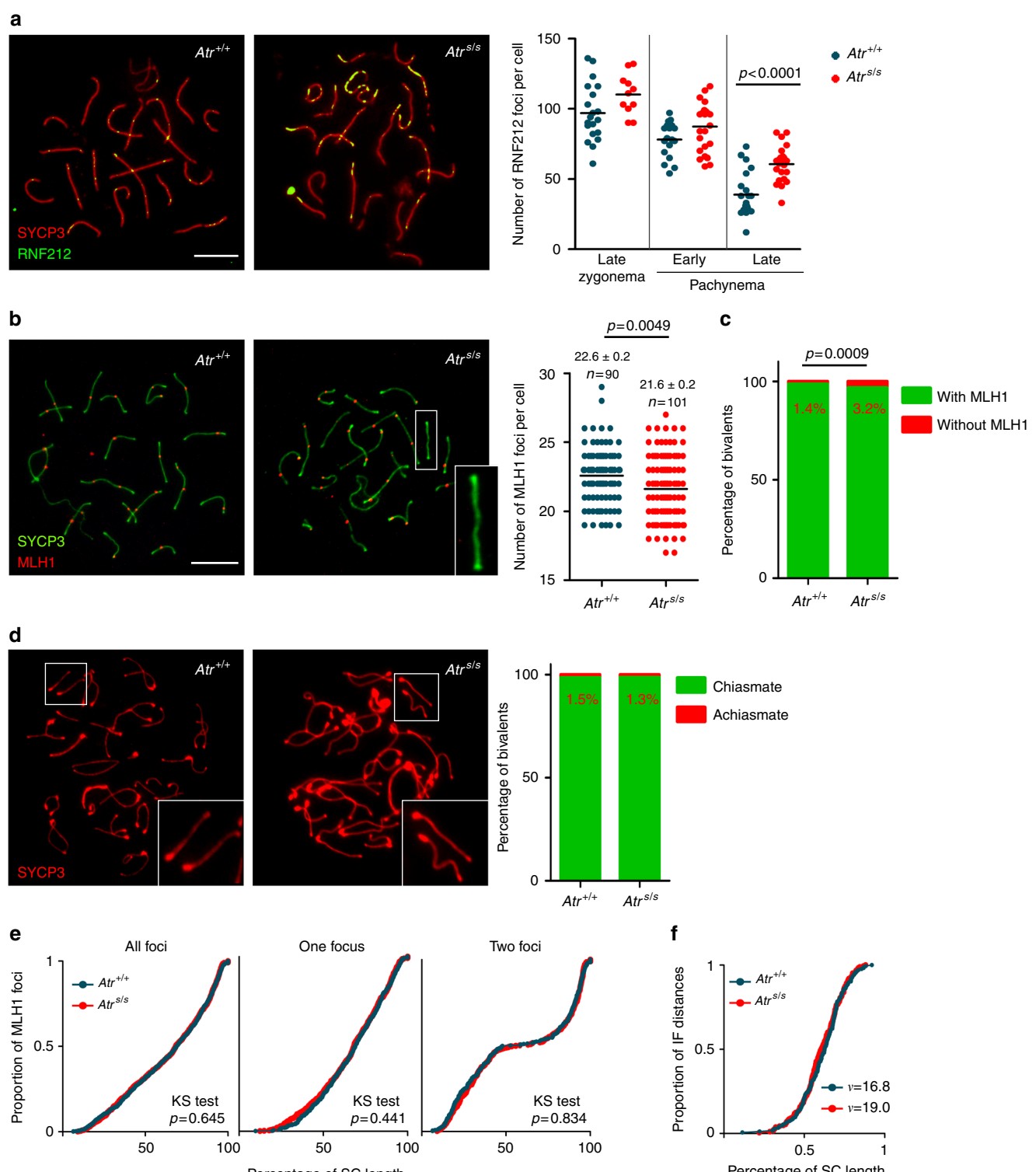

molecules from circulating blood into the seminiferous tubules[38]. Thus, we cultured fragments of neonatal testes in media containing AZ20. We hypothesized that this strategy could surmount the blood–testis barrier, thereby increasing exposure of spermatocytes to AZ20 and better inhibiting ATR function.

We cultured testis fragments isolated from 5 days post partum (dpp) wild-type mice. Because ATR inhibition can interfere with DNA replication[39], fragments were initially cultured without treatment for 7 days to allow entry into meiosis. Then, for a further 7 days, fragments were cultured either without treatment or in the presence of different doses of AZ20 (0.2, 1, or 5 μM) or equivalent volumes of DMSO (DMSO1, DMSO2, or DMSO3).

We used SYCP3 to assess meiotic prophase initiation, progression, and synapsis. At day zero (D0, i.e., prior to culture), only 1% of testis cells were SYCP3-positive and were thus judged to be spermatocytes (Fig. 5a, b). After the initial 7 days of culture (D7), 13% of testis cells were spermatocytes, of which 75% were at leptonema. After 2 weeks of culture (D14), 16% of testis cells from untreated samples showed SYCP3 staining. Importantly, 25% of these D14 spermatocytes displayed complete synapsis consistent with having reached pachynema. A low dose of AZ20 had no effect on meiotic progression: at D14, proportions of spermatocytes at different prophase stages were indistinguishable between 0.2 μM AZ20-treated, DMSO1-treated, and untreated samples (Fig. 5a–d). However, at higher doses (1 or 5 μM), cultured samples exhibited a decreased fraction of SYCP3-positive cells compared with DMSO-treated controls, suggesting that some spermatocytes had died between D7 and D14. We did not detect any spermatocytes with complete synapsis or with a sex body (Fig. 5c–e, Supplementary Table 3).

We used 5 μM AZ20-treated samples to evaluate whether recombination defects observed in the in vivo models were recapitulated in vitro. Whereas untreated and DMSO3 samples exhibited similar RAD51 focus dynamics, dramatically fewer foci were observed in 5 μM AZ20-treated samples (Fig. 5f, g, Supplementary Table 4). A severe effect on DMC1 focus formation was likewise seen (Fig. 5h). Similar to the in vivo models, RPA focus counts at early leptonema were higher from 5 μM AZ20-treated samples than in controls (Fig. 5i, Supplementary Table 4). DMSO treatment alone appeared to cause modest changes, as fewer RPA foci were observed in late leptotene and late zygotene spermatocytes in DMSO3 samples compared with untreated, possibly due to DMSO toxicity (Fig. 5i). Nonetheless, AZ20-treated samples behaved differently from DMSO3-treated samples at all prophase stages. These data show that inhibition of ATR function alters RAD51 and DMC1 loading.

**CHK1 function is required for proper RAD51 loading.** Checkpoint kinase 1 (CHK1) is the major effector of the ATR signaling pathway[3]. In somatic cells ATR activates CHK1, which

interacts with and phosphorylates RAD51 to promote repair of DNA lesions[3,40]. To investigate the effect of CHK1 perturbation on meiotic recombination, we targeted CHK1 protein in vitro with two specific inhibitors, PF-477736[41] and LY2603618[42] (referred to as CHK1i). As above, we cultured 5 dpp testis fragments for 7 days before subjecting them to CHK1i for an additional 7 days.

As assessed by SYCP3 immunostaining at D14, CHK1i-treated samples displayed fewer SYCP3-positive spermatocytes than DMSO-treated controls and no spermatocytes that had completed synapsis (Fig. 5c). Moreover, RAD51 focus numbers were substantially reduced, indicating that RAD51 loading onto chromosome axes was severely compromised (Fig. 5j, k). We conclude that disruption of the ATR-CHK1 pathway affects loading of RAD51 onto resected DSBs and restricts progression through meiotic prophase.

**ATR promotes SC elongation.** The absence of spermatocytes with complete SC formation after in vitro inhibition of ATR or CHK1 (Fig. 5c–e, Supplementary Table 3) suggested that ATR may participate in SC formation and/or elongation. We used a Spo11 mutant to study how in vivo inhibition of ATR affected SC formation and/or elongation in the absence of meiotic recombination[43]. Spo11[−/−] spermatocytes fail to form DSBs but still form stretches of SC, so SC formation is independent of meiotic recombination in this genetic background. AZ20-treated Spo11[−/−] spermatocytes displayed the same number of SC stretches per cell as untreated Spo11[−/−] cells (Supplementary Fig. 2a, b). However, the average length of these fragments was shorter in AZ20-treated spermatocytes (Supplementary Fig. 2a, c). These findings suggest that wild-type levels of ATR function are not required for SC initiation but play an important role in SC elongation.

**Accumulation of recombination markers on unsynapsed axes.** In yeast, Mec1 regulates meiotic recombination partner choice by phosphorylating Hop1[44,45], the ortholog of mouse HORMA domain-containing protein HORMAD1. HORMAD1 is involved in several meiotic processes, localizes to chromosome axes at the onset of meiosis and becomes phosphorylated during meiotic progression[46,47] (discussed further below). In mouse, unsynapsed chromosomes tend to accumulate recombination markers during zygonema[48–51], implying that DSBs are continuously formed and processed in unsynapsed regions until the homolog is engaged for repair. Because ATR decorates unsynapsed axes in mice[5], where HORMAD1 accumulates, we asked whether ATR plays a role in DSB formation and early processing specifically in unsynapsed regions.

The X chromosome normally remains completely unsynapsed throughout zygonema[40], so it is a useful model for examining the kinetics of recombination markers on unsynapsed axes[48]. We

**Fig. 3** Late recombination markers in wild type and Atr[S/S] spermatocytes. **a** Left panels, representative images of late pachytene spermatocytes immunostained against SYCP3 and RNF212. Scale bar = 10 μm and applies to both images. Right panel, quantification of RNF212 foci. Horizontal black lines denote the means; the p value is from t test, and comparisons without p values stated were not significant (p > 0.05). **b** Left panels, representative images of pachytene spermatocytes immunostained against SYCP3 and MLH1. Scale bar = 10 μm and applies to both images. A bivalent lacking an MLH1 focus in the Atr[S/S] spermatocyte (white box) is magnified in the inset. Right panel, quantification of autosomal MLH1 foci at pachynema. Horizontal black lines denote the means. Atr[S/S] and control were compared by Mann–Whitney test. **c** Proportion of bivalents without an MLH1 focus. N = 1520 (mutant) or 1653 (control); p value is from Fisher's exact test. **d** Left, representative images of diplotene spermatocytes immunostained against SYCP3. Scale bar = 10 μm. Achiasmate bivalents (white boxes) are magnified in the insets. Right, percentage of achiasmate bivalents at late diplonema from Atr[+/+] (N = 798) and Atr[S/S] (N = 912) spermatocytes (p = 0.84, Fisher's exact test). **e** Cumulative frequency plots comparing MLH1 focus distribution along autosomal bivalents from pachytene Atr[+/+] and Atr[S/S] spermatocytes. MLH1 focus distributions were similar between Atr[+/+] (N = 924) and Atr[S/S] (N = 932). Distributions in SCs presenting one MLH1 focus from Atr[+/+] (N = 581) and Atr[S/S] (N = 601) and in SCs presenting two MLH1 foci from Atr[+/+] (N = 350) and Atr[S/S] (N = 369) spermatocytes were also indistinguishable (Kolgomorov–Smirnov (KS) test). **f** MLH1 interfocal distances (as a percentage of SC length) measured on SCs containing more than one MLH1 focus. Data were fitted to the gamma distribution to measure the strength of interference denoted by the shape parameter ($\nu$)[34]

scored RAD51 and RPA focus density on the X chromosome at early and late zygonema in wild-type and $Atr^{S/S}$ spermatocytes (Fig. 6a–d). The densities of RAD51 and RPA foci in wild type were higher in late zygonema than early zygonema ($p = 0.0005$ for RAD51 and $p = 0.023$ for RPA, $t$ test) (Fig. 6a–d), as previously reported[48]. In $Atr^{S/S}$ spermatocytes in contrast, neither RAD51 focus density, which was substantially lower at both substages, nor RPA focus density rose significantly as zygonema progressed ($p > 0.05$, $t$ test) (Fig. 6a–d).

To corroborate these findings, we asked whether culturing testis fragments in 5 μM AZ20 also curtailed accumulation of recombination markers on unsynapsed chromosomes (Fig. 6e). Indeed, AZ20-treated samples exhibited a significant decrease in the density of RPA foci on unsynapsed axes as zygonema progressed ($p = 0.0004$, $t$ test). Similar to our earlier observations for total RPA counts, DMSO treatment affected RPA focus density on unsynapsed axes, but AZ20 treatment further reduced this density. These findings indicate that ATR contributes to the

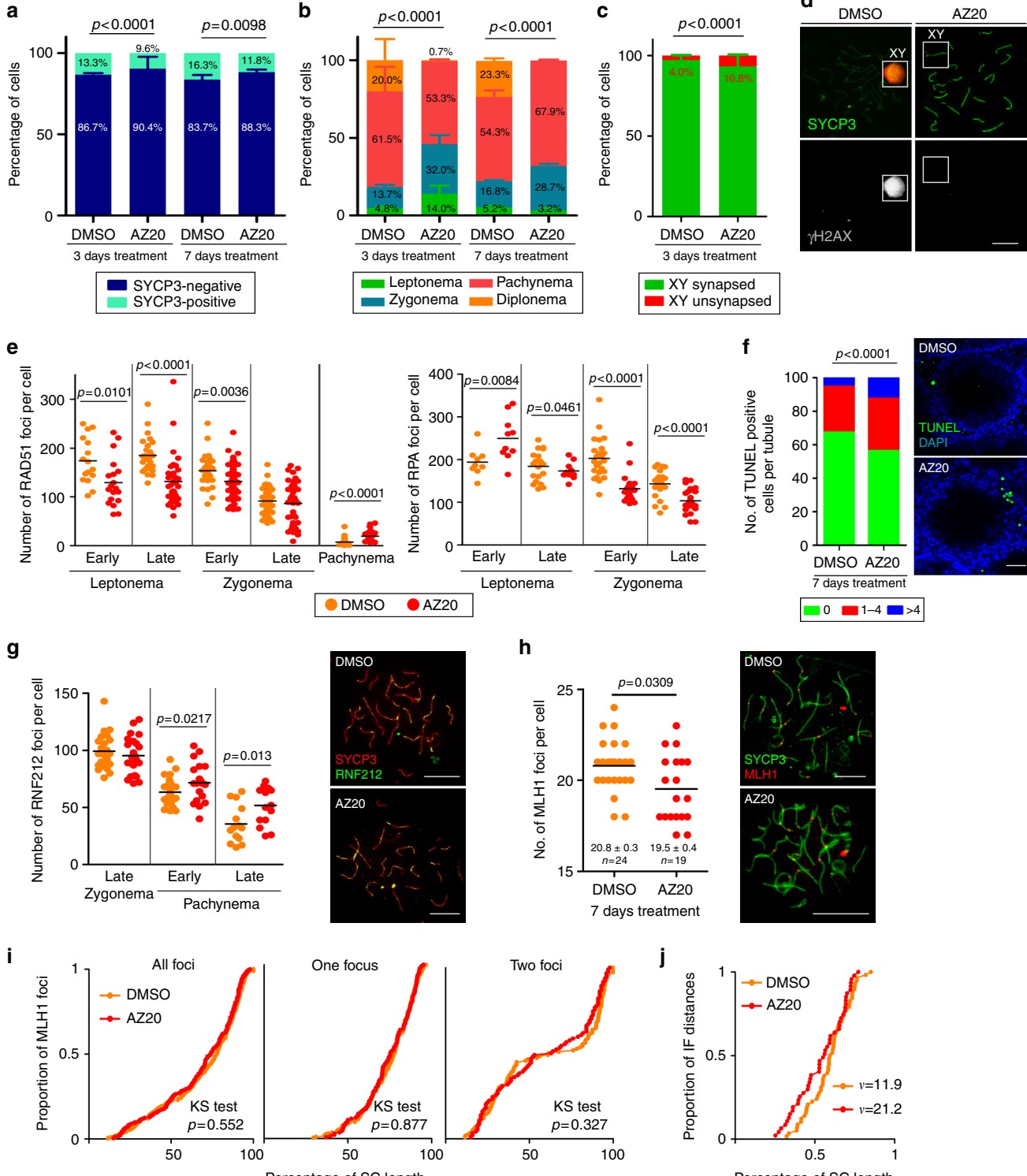

normal patterns of accumulation of recombination markers on unsynapsed chromosome axes.

## Discussion

In this study, we used a combination of genetic and pharmacological tools to examine the function of ATR in meiotic recombination. This strategy is a useful approach for studying the meiotic role(s) of a protein whose null mutation is embryonically lethal. Although conditional models can also address this issue, the use of specific inhibitors is advantageous because mice of desired genotypes are more readily obtained when studying a single locus. This aspect significantly reduces the number of mice required to perform the study.

An accompanying study using conditional *Atr* mutant mice corroborates our results by reporting the role of ATR in promoting homologous chromosome synapsis, SC elongation, RAD51 and DMC1 loading onto DSBs, and maintenance of recombination foci on unsynapsed axes[52]. However, this mutant mouse displays meiotic arrest at mid pachynema that prevents studying ATR functions past this point. In this sense, the use of a hypomorphic mutation and a pharmacological strategy has proven effective to investigate ATR functions in some meiotic processes, like crossover formation and distribution, that are obscured by arrest of prophase progression. Thus, these two strategies complement one another.

Using our approach, ATR function could be inhibited to different degrees, as determined by meiotic prophase progression and sex-body formation. ATR is known to be required to form the sex body[5]. Upon in vitro administration of AZ20, spermatocytes were blocked by zygonema (i.e., prior to sex-body formation), so we infer that this condition delivered the strongest inhibition. In vivo administration of AZ20 allowed accumulation of pachytene spermatocytes, thus inhibition was not as strong as in the in vitro setting. Finally, $Atr^{S/S}$ spermatocytes displayed grossly normal meiotic progression and nearly wild-type levels of sex-body formation. Of note, markers of other meiotic processes (e.g., RAD51 focus counts and XY synapsis) were also more perturbed in AZ20-treated samples than in $Atr^{S/S}$ mice.

Our data show that ATR and its effector CHK1 play a critical role in the loading of RAD51 and DMC1 onto chromatin. While RAD51 focus numbers were reduced in $Atr^{S/S}$ spermatocytes relative to wild type only at early leptonema, DMC1 focus numbers were significantly lower from early leptonema through late zygonema. As expected from previous studies performed in mammals and other organisms[53,54], the different effects on the kinetics of RAD51 and DMC1 foci corroborate that these recombinases are regulated differently. Distinct regulation may be

attributable to several factors. First, while the protein sequences of mouse RAD51 and DMC1 are very similar (53.3% identical and 64% similar), they differ at a residue that may be critical for RAD51 activity. In mammalian somatic cells, RAD51 Threonine 309 is phosphorylated by CHK1 upon activation of ATR by RPA-coated DNA structures, and this post-translational modification promotes RAD51 focus assembly[40]. It has been suggested that CHK1-dependent RAD51 phosphorylation is required for RAD51 to displace RPA from resected DSB ends and initiate homology search[40]. The paralogous residue in DMC1 is Leucine 310, hence it cannot be similarly regulated. Second, the current favored model of RAD51 and DMC1 loading onto resected DSBs in meiosis proposes that RAD51 loading is a prerequisite for the incorporation of DMC1[53]. Thus, this interdependency could also account for the more severe effect on DMC1 focus formation observed in ATR-defective spermatocytes.

Aberrant loading and kinetics of RAD51 and DMC1 may have implications for homologous chromosome pairing and synapsis. Cytologically, the phenotype of ATR-deficient spermatocytes (altered kinetics of recombination markers, reduced number of MLH1 foci at pachynema, increased proportion of spermatocytes at zygonema, and defects in sex-chromosome synapsis) is consistent with impaired meiotic recombination. However, our data in the DSB-deficient $Spo11^{-/-}$ background indicate that ATR can also promote SC extension independently of its role in meiotic recombination. Several proteins of the chromosome axis, including HORMAD1, are phosphorylated during meiotic prophase[55]. HORMAD1 is required for normal numbers of meiotic DSBs, proper SC formation and recruitment of ATR onto unsynapsed chromosome axes[47]. At the onset of meiotic prophase HORMAD1 is localized at the chromosome axes[46]. As a result of DSB formation, HORMAD1 is phosphorylated, presumably by ATR[5,55], and when homologous chromosomes synapse, HORMAD1 is displaced from the axes. Thus, we hypothesize that the role of ATR in SC elongation may be mediated by HORMAD1. Indeed, the yeast HORMAD1 ortholog Hop1 is phosphorylated by Mec1 and Tel1 (the orthologs of ATR and ATM), and this phosphorylation promotes SC formation[44].

In budding yeast, Mec1 participates in two additional key aspects of meiotic recombination. First, it biases recombination toward the homologous chromosome over the sister chromatid as the preferred template for meiotic DSB repair[45]. Second, it indirectly promotes DSB formation during meiotic prophase by inhibiting the transcription factor Ndt80, which is required to exit pachynema and thereby close the cell cycle window during which Spo11 forms DSBs[56]. In yeast, inter-sister meiotic recombination is over three times faster than inter-homolog recombination[44]. If ATR participates similarly in the implementation of inter-

**Fig. 4** In vivo inhibition of ATR affects prophase progression and recombination markers. **a** Percentage of SYCP3-positive cells from mouse testes treated 3 or 7 days with DMSO ($N = 2057$ and $N = 986$ cells, respectively) or AZ20 ($N = 2063$ and $N = 936$ cells, respectively). Two mice per condition were analyzed. *P* value is from Fisher's exact test. **b** Percentage of spermatocytes at different prophase stages in 3 and 7 days DMSO- ($N = 2035$ and $N = 600$) and AZ20-treated mice ($N = 2091$ and $N = 600$ cells, respectively). Two mice per condition were analyzed. *P* value is from G test. **c** Percentage of pachytene cells exhibiting unsynapsed X and Y chromosomes in DMSO- ($N = 472$) and AZ20-treated spermatocytes ($N = 426$). Two mice per condition were analyzed. *P* value is from Fisher's exact test. **d** Representative images of DMSO- and AZ20-treated pachytene spermatocytes stained for SYCP3 and γH2AX. Note the presence of a sex body over the X and Y chromosomes in the control, but not in the AZ20-treated cell. **e** RAD51 and RPA foci per spermatocyte at the indicated stages in DMSO- and AZ20-treated mice. Horizontal lines denote the means. *P* values are from *t* tests. **f** Proportion of tubule sections with 0, 1–4, or >4 TUNEL-positive cells from mice treated with DMSO or AZ20. *P* value is from *t* test. Scale bar = 40 μm. **g** RNF212 foci in pachytene spermatocytes after 7 days of DMSO or AZ20 treatment. Horizontal black lines denote the means. *P* value is from *t* test. Images show pachytene spermatocytes immunostained for SYCP3 and RNF212. Scale bars = 10 μm. **h** Autosomal MLH1 foci in pachytene spermatocytes after 7 days of DMSO or AZ20 treatment. Horizontal black lines denote the means. *P* value is from a Mann–Whitney test. Images show pachytene spermatocytes immunostained for SYCP3 and MLH1. Scale bars = 10 μm. **i** Cumulative frequency plots comparing MLH1 focus distribution along autosomal bivalents from pachytene spermatocytes after 7 days of DMSO or AZ20 treatment. MLH1 focus distribution along all autosomal bivalents was similar between DMSO ($N = 320$) and AZ20 ($N = 233$) treatments. In SCs presenting one MLH1 focus from DMSO- ($N = 204$) and AZ20- ($N = 138$) treated mice or two MLH1 foci from DMSO- ($N = 116$) and AZ20- ($N = 94$) treated mice, MLH1 focus location was also indistinguishable (KS test). **j** MLH1 interfocal distances expressed as a percentage of SC length from autosomal bivalents containing two MLH1 foci. Shape parameters (υ) from gamma distribution are shown[34]

homolog bias in mouse, ATR-deficient spermatocytes might repair DSBs more rapidly by recombination between sister chromatids. This model could explain the observed reduction in recombination marker density on unsynapsed axes at zygonema. In support of this model, the reduction in RPA foci at zygonema in $Atr^{S/S}$ spermatocytes despite globally unaltered DSB numbers could reflect a more rapid turnover of recombination markers by

inter-sister repair. However, our data also provide results that seem inconsistent with this interpretation. The increase in recombination markers (γH2AX, RAD51, and DMC1) on the synapsed axes of pachytene-stage $Atr^{S/S}$ spermatocytes appears difficult to reconcile with faster repair by inter-sister recombination. However, it is also plausible that ATR may play another role in DSB repair from mid/late pachynema onward. It has been

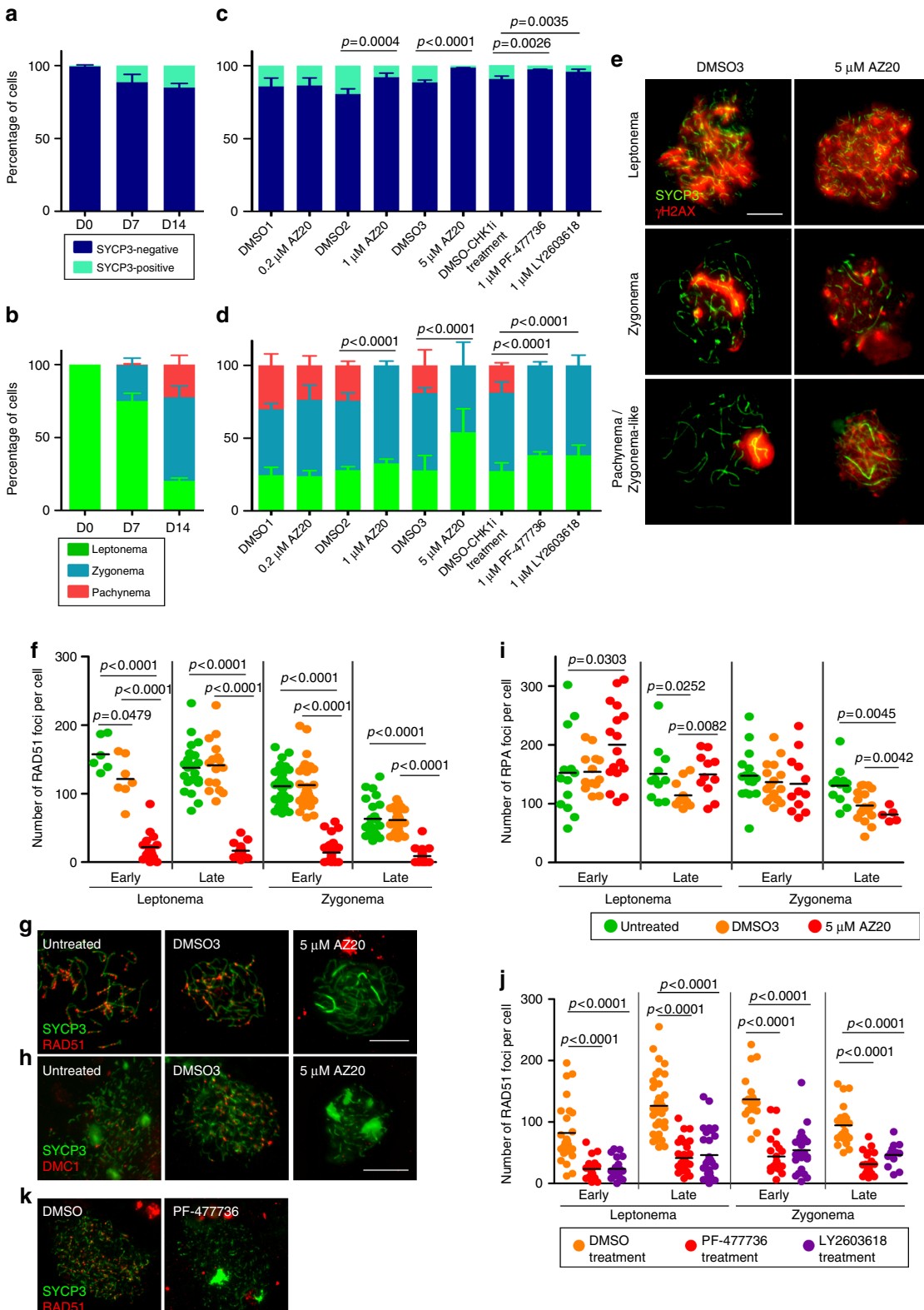

proposed that DSB repair processes other than canonical meiotic recombination may be active from mid pachynema onward[19], including non-homologous end joining (NHEJ)[57]. Spermatocytes from mice deficient for KU70, which is required for NHEJ, present more γH2AX patches at late pachynema and diplonema than control cells, suggesting that NHEJ participates in DSB repair at late meiotic prophase[57]. Thus, the accumulation of γH2AX patches observed at mid/late pachynema and diplonema in Seckel mice could also reflect that ATR is required to activate alternative DSB repair pathway(s) at later stages of meiotic prophase.

Finally, the presence of normal numbers of achiasmate bivalents at diplonema in Seckel mouse spermatocytes suggests that ATR function in that context is sufficient for normal levels of crossover formation. We did not observe enough in vivo AZ20-treated diplotene spermatocytes to study chiasma formation when ATR function was further reduced. Nonetheless, our data show that ATR function is required for the proper number of MLH1 foci at pachynema.

In summary, our data show that the ATR-CHK1 pathway is required for the completion of meiotic recombination in mammalian spermatocytes. RAD51 and DMC1 kinetics are altered when ATR or CHK1 function is attenuated, suggesting a role for this pathway in loading RAD51 and DMC1 onto resected DSBs. Furthermore, deficiency in ATR function affects SC formation and the accumulation of recombination markers along unsynapsed axes during zygonema. Thus, reduction in ATR function may impair the ability of homologous chromosomes to pair and synapse during meiotic prophase. Finally, our finding that in vivo exposure to AZ20 causes meiotic prophase arrest is relevant when considering the use of ATR inhibitors in cancer therapy, since meiotic arrest may result in infertility. Further studies should be performed to determine how long this arrest lasts after exposure to the drug.

## Methods

**Mice and genotyping.** The *Atr Seckel (AtrS)* and *Spo11* alleles were generated previously[14,43]. Genotyping was performed by PCR of tail tip DNA using previously designed primers[14,43]. The *AtrS* allele was maintained on a C57BV6-129/Sv mixed genetic background. Experiments were performed using two or more mutant or treated animals and compared with littermate control mice. When appropriate littermate controls were unavailable, control animals were obtained from other litters of the same matings and at the same age. All animals were killed using CO$_2$ euthanization methods. Experimental procedures conform to the protocol CEEAH 1091 (DAAM6395) approved by the Ethics Committee for Animal Experimentation of the Universitat Autònoma de Barcelona and the Catalan Government.

**RNA expression analysis.** RNA from mouse testes was extracted using the RNeasy Plus Mini kit (Qiagen). The SuperScript III One-Step RT-PCR Taq High Fidelity (Invitrogen) was used to synthesize cDNA and amplify *Atr* transcripts using previously designed primers[16].

**Cytology, immunostaining, and FISH.** Testes were dissected from mice at 2–3 months of age and were processed for cytology and histology, as previously described[50,58]. Briefly, for surface spreading procedures, the tunica albuginea was removed and seminiferous tubules were released and minced to obtain a single-cell suspension. Cells were treated with a hypotonic solution (0.1 M sucrose) and fixed in 1% paraformaldehyde in H$_2$O with 0.1% Triton X-100 for 2 h. Slides were washed in 0.4% Kodak Photo-Flo and air-dried. Immunostaining of surface spreads was performed using standard methods[37]. The following primary antibodies were used: rabbit anti-SYCP3 (Abcam) 1:200 dilution, mouse anti-SYCP3 (Abcam) 1:200 dilution, mouse anti-SYCP1 (Abcam) 1:400 dilution, mouse anti-γH2AX (Millipore) 1:200 dilution, guinea pig anti-H1T (kindly donated by Dr. Mary Ann Handel) 1:500 dilution, rabbit anti-ATR (Calbiochem) 1:100 dilution, mouse anti-SUMO-1 (Life Technologies) 1:100 dilution, sheep anti-MDC1 (AbdSerotech) 1:100 dilution, rabbit anti-RAD51 (Calbiochem) 1:100 dilution, rabbit anti-RPA (kindly donated by Dr. Edita Marcon) 1:100 dilution, and mouse anti-MLH1 (BD Bioscience) 1:50 dilution, goat anti-RNF212 (kindly donated by Dr. N. Hunter) 1:150 dilution, rabbit anti-p-S271-HORMAD2 (kindly donated by Dr. A. Toth) 1:250 dilution, rabbit anti-p-S33-RPA32 (Bethyl) 1:100 dillution, rabbit anti-p-S317-CHK1 (Cell Signaling) 1:25 dillution. These antibodies have been routinely used to study meiotic progression in mammals[50,58]. Combined immuno-fluorescence/FISH was performed using a BAC probe RP24-204O18 (CHORI BACPAC library) labeled with digoxigenin to detect the X-linked *Scml2* gene. After immunostaining, slides were treated with a pre-warmed denaturation solution (70% formamide in 2% SSC, pH 7.2–7.4) for 20 min at 74 °C, incubated in a humid chamber for 2.5 h at 65 °C with 100 μl of 1 M sodium thiocyanate, and immediately immersed in denaturation solution for 20 min at 74 °C. Slides were dehydrated in a series of ethanol solutions. A volume of 5 μl of probe was mixed with one volume of hybridization buffer (4 × SSC, 50% dextran sulfate, 2 mg/ml BSA, 2 mM vanadyl ribonucleoside), denatured for 10 min at 75 °C and reannealed for 10 min at 37 °C. Slides were incubated with hybridization mix for 72 h at 37 °C. Slide post-hybridization treatments consisted of three washes for 5 min in washing solution (50% formamide, 2 × SSC pH 7.2–7.4) at 45 °C, three washes for 5 min in 2 × SSC at 45 °C, one immersion in 4 × SSC, 0.1% Tween 20 at room temperature, one incubation for 30 min with blocking buffer (4 × SCC, 4 mg/ml BSA, 0.1% Tween-20) at 37 °C, one incubation for 1 h with 30 μl of digoxigenin detection solution (10% Anti-Digoxigenin-Fluorescein (ApopTag Plus Fluorescein In Situ Apoptosis Detection kit (Millipore)) in 4 × SSC, 0.1% Tween 20) at 37 °C, and three washes for 5 min in 4 × SSC, 0.1% Tween 20 at room temperature. DAPI (4′,6-diamidino-2-phenylindole) (Sigma) was used to stain the DNA.

**RNA-FISH and immunofluorescence.** RNA FISH was carried out using BAC probe bMQ-372M23 (Mouse bMQ BAC library) labeled with digoxigenin to detect the *Zfx* gene, as previously described[59]. Briefly, BAC-containing bacteria were grown in an overnight LB-Chloramphenicol culture at 37 °C and a standard miniprep method was used to isolate BAC DNA. 2 μg of BAC DNA were labeled using DIG-Nick Translation Mix (Roche) and precipitated with Cot-1 DNA (Invitrogen) and salmon sperm DNA (Stratagene). Mouse testes were minced, then cells were treated with CSK buffer (100 mM NaCl, 300 mM sucrose, 3 mM MgCl2, 10 mM PIPES, 0.5% Triton X-100, 2 mM vanadyl ribonucleoside (New England Biolabs)) and fixed in 4% paraformaldehyde in PBS, and slides were dehydrated in a series of ethanol solutions. 30 μl of probe was denatured for 10 min at 80 °C and reannealed for 30 min at 37 °C. Slides were incubated overnight at 37 °C. Slide post-hybridization treatments consisted of three washes for 5 min in 50% formamide, 2 × SSC pH 7.2–7.4 solution at 45 °C, three washes for 5 min in 2 × SSC at 45 °C, one immersion in 4 × SSC, 0.1% Tween 20 at room temperature, one incubation for 30 min with blocking buffer (4 × SCC, 4 mg/ml BSA, 0.1% Tween-20) at 37 °C, and one incubation for 1 h with anti-digoxigenin-FITC (1:10, Millipore). RNA FISH was then followed by immunostaining with an anti-HORMAD1 antibody (Abcam) at 1:50 dilution. Cells were examined on an Olympus IX70 inverted microscope. Images were captured using a computer-assisted (DeltaVision) CCD camera (Photometrics). Pachytene cells were defined based on continuous HORMAD1 staining along the X- and Y-chromosome axes.

**SPO11-oligonucleotide complex detection and western blotting.** Testis extract preparation, immunoprecipitation, SPO11-oligonucleotide detection and western blot analysis were performed essentially as previously described[60]. Briefly, SPO11-oligonucleotide complexes and free SPO11 were isolated from testis lysates by two

**Fig. 5** In vitro inhibition of ATR and CHK1 block recombination and prophase progression. **a** Percentage of SYCP3-positive cells in untreated testis fragments at 0, 7, and 14 days of culture. Columns and lines indicate the mean and SD from four replicates. **b** Mean percentage of spermatocytes at the indicated stages of prophase in untreated testis fragments from two replicates. **c** Proportion of SYCP3-positive cells at D14 in testis fragments treated with the indicated dosages of AZ20, PF-477736, or LY2603618, and their respective DMSO controls. Data obtained from three replicates per each condition. *P* values are from Fisher's exact tests. **d** Proportion of spermatocytes at different stages of meiotic prophase at D14 of culture. Data obtained from two replicates per each condition. *P* values are from G tests. **e** Representative images of spermatocytes from testes treated with DMSO3 or 5 μM AZ20 showing progression of meiotic prophase, followed by staining for SYCP3 and γH2AX. **f** RAD51 foci at the indicated stages from D14 cultures (Color code key is in panel **i**). In this and other graphs of focus counts, horizontal lines denote the means and *p* values are from pairwise *t* tests. **g**, **h** Representative spermatocytes from cultured samples stained for SYCP3 and either RAD51 (**g**) or DMC1 (**h**). **i** RPA foci in cultured spermatocytes. **j** RAD51 foci per spermatocyte from control and PF-477736- or LY2603618-treated samples. **k** Representative images of spermatocytes from D14 cultures treated with DMSO or 1 μM PF-477736 and immunostained for SYCP3 and RAD51. Data presented for each culture condition correspond to at least two experiments performed using different samples. Scale bars in all micrographs represent 10 μm

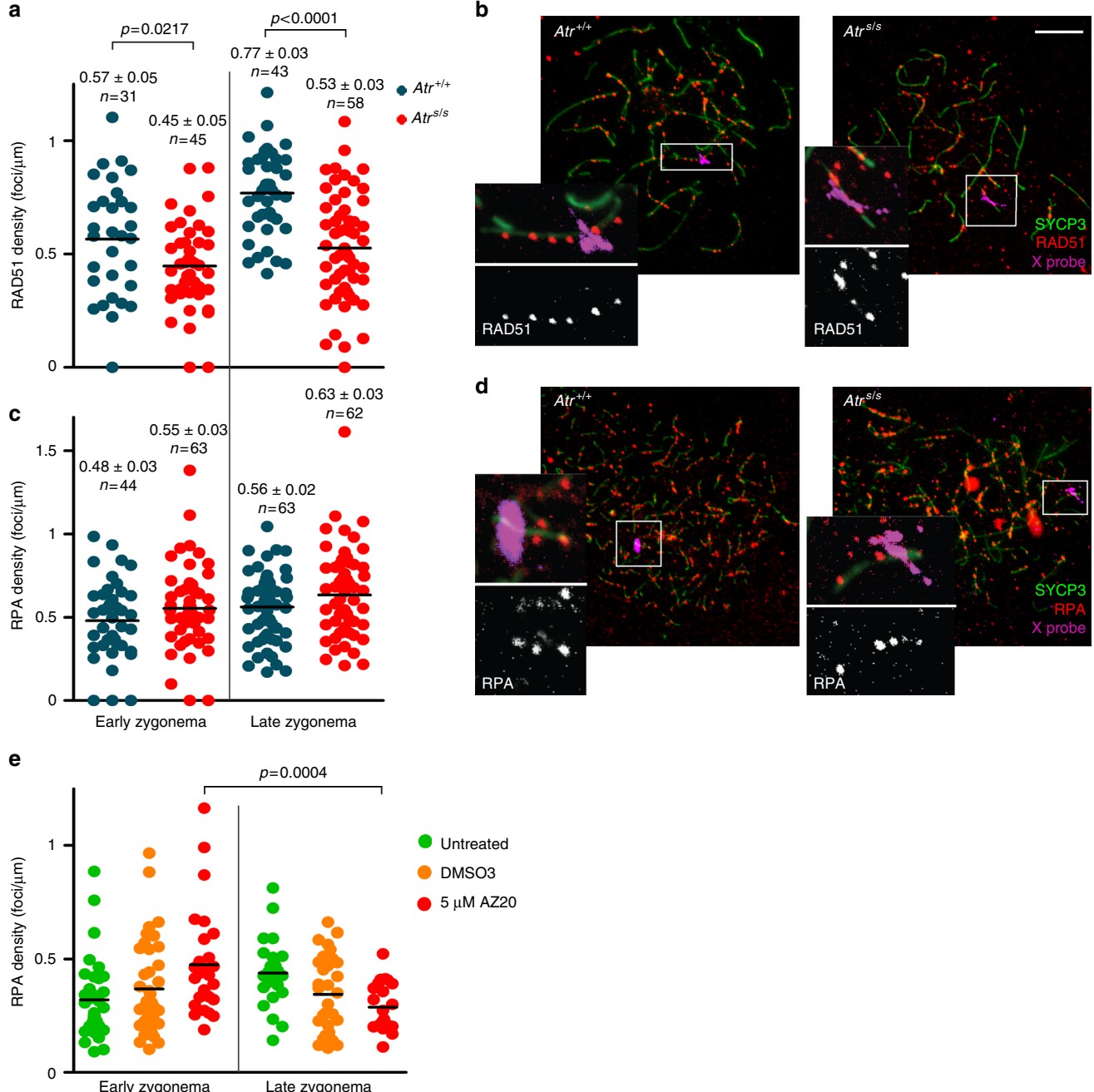

**Fig. 6** ATR function is needed for normal levels of recombination markers on unsynapsed axes. **a** RAD51 focus density along the entire measurable X-chromosome axis. Horizontal black lines denote the means. *P* values are from *t* tests. **b** Representative images of late zygotene spermatocytes. Images show overlays of immunofluorescence against SYCP3 and RAD51 with fluorescence in situ hybridization (FISH) for the X-PAR boundary. Scale bar = 10 μm. Inset images show RAD51 foci on the measurable X-chromosome axis. **c** RPA focus density on the measurable X chromosome. **d** Images of early zygotene spermatocytes showing immunofluorescence against SYCP3 and RPA, overlaid with FISH for X-PAR boundary. Inset images show RPA foci on the measurable X-chromosome axis. **e** RPA focus numbers present on unsynapsed axes at D14 from spermatocytes cultured in vitro

rounds of immunoprecipitation with an anti-SPO11 monoclonal antibody (Spo11-180) on Protein A-agarose beads (Sigma). SPO11-oligonucleotide complexes were labeled with [α-$^{32}$P] dCTP using terminal deoxynucleotidyl transferase (Fisher), released by boiling in Laemmli buffer and fractionated by SDS-PAGE. The electrophoresed products were transferred onto polyvinylidene fluoride (PVDF) membrane. Radiolabeled species were detected using Fuji phosphor screens and quantified with ImageGauge software. The membrane was probed with the SPO11 antibody at 1:2000 dilution.

For western blot analysis, wild type and Seckel mouse testis protein extracts were obtained by mincing the tissues in RIPA buffer and boiling them for 10 min. After centrifugation, supernatants were run in a 4–20% gradient SDS-PAGE. Transfer to a nitrocellulose membrane was performed using a Trans-Blot Turbo System (Bio Rad).

The membrane was probed with ATR antibody (Abcam, 1:15000 dilution) and tubulin antibody (Abcam, 1:5000 dilution).

**In vivo drug administration**. AZ20 (Selleckchem) compound dissolved in 10% DMSO/40% propylene glycol/50% water was administered orally to adult wild-type males at a single daily dose of 50 mg/kg for 3 or 7 consecutive days or to *Spo11*$^{-/-}$ males for 7 days. Control animals were administrated with the same volume of vehicle solution. Allocation of treated and control animals to each group was performed randomly. Animals were killed 24 h after the treatment ended.

**Neonatal testis organ culture**. Five dpp mouse testes were cultured as previously described[61]. Briefly, 1.5% agarose gel cubes were placed in a six-well plate and

pre-soaked overnight in culture medium (10% KSR (Invitrogen), 1% Antibiotic-Antimycotic 100 × (Gibco), 0.37% sodium bicarbonate in α-MEM (Invitrogen) in a culture incubator supplied with 5% $CO_2$ in air and maintained at 34 °C. On the day of the culture, medium from the six-well plates was removed and replaced with fresh culture medium. Testis fragments from 5 dpp mice were placed on top of the agarose cubes and incubated for 7 days with 5% $CO_2$ in air and 34 °C. Samples were then treated with 0.2, 1, or 5 µM AZ20, 1 µM PF-477736 or 1 µM LY2603618 (all drugs from Selleckchem) dissolved in DMSO (Sigma). Control samples were treated with equivalent volumes of DMSO used for each inhibitor concentration. Testis fragments were incubated with 5% $CO_2$ in air at 34 °C for 7 more days. Meiotic spreads from cultured testis fragments were performed at day 0, 7, or 14 of culture. Surface spreads were immunostained to analyze meiotic progression and recombination.

**SC length, MLH1 distribution, and recombination focus density**. Chromosomes axis length, SC length, and recombination focus position along the axis/SCs were recorded using MicroMeasure, as previously published[20]. Briefly, positions of MLH1 foci were recorded as the distance from each focus to the centromeric end of the chromosome, identified by the brighter DAPI staining of pericentromeric heterochromatin. MLH1 positions were expressed as a percentage of the total chromosome (or SC) length and are presented as cumulative graphs (Figs. 3e and 4i). Similarly, to measure cytological interference, the distances between adjacent MLH1 foci (IF, interfocal distances) from those SCs displaying more than one MLH1 focus were expressed as a percentage of the SC length and presented as cumulative graphs (Figs. 3f and 4j). Recombination focus density was obtained by dividing the number of recombination foci by the chromosome axis length in micrometers.

**Statistical analysis**. Student's $t$ tests, Mann–Whitney tests and one-way ANOVA tests were performed using GraphPad Prism software and/or GraphPad Quick-Calcs online resource (http://www.graphpad.com/quickcalcs/).

**Image processing and data analysis**. Microscopy analysis was performed blindly whenever possible. Nonetheless, mutant and treated samples displayed a particular phenotype (see Results section) that allowed their identification. Image acquisition was performed using a Zeiss Axioskop microscope connected with a ProgRes Jenoptik camera. Images were captured using ProgRes CapturePro software and were processed using Photoshop and ImageJ to quantify fluorescence intensity and chromosome axes length and/or MicroMeasure version 3.3 to analyze MLH1 focus position and axis length, as previously described[20].

**Data availability**. All relevant data are presented in the figures or supplementary figures and tables and are available from the authors upon request.

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

## Acknowledgements

We thank M. A. Handel (The Jackson Laboratory, Bar Harbor, USA) for the anti-H1T antibody; E. Marcon for the anti-RPA antibody (University of Toronto, Canada); A. Toth for the anti-pHORMAD2 antibody (U. Dresden, Germany) and N. Hunter for the anti-RNF212 antibody (UC Davis, USA); J. Turner (National Institute for Medical Research, London, UK) for assistance in the RNA-FISH experiments, for the X chromosome probe, for providing $Atr^{FL/-}$ testis samples and for sharing unpublished data; L. Kauppi (University of Helsinki, Finland) for providing us with protocols for the testis cultures; and members of the Roig lab and the Spanish Ministerio de Ciencia e Innovación-funded Network of Spanish groups working on Meiosis (MeioNet, BFU2015-71786-REDT) and Enrique Martínez Pérez (Imperial College, London, UK) for helpful discussions. M.M.O. was supported by a FPI fellowship from the Ministerio de Ciencia e Innovación (BES-2011-045381). J.L. was supported in part by American Cancer Society post-doctoral fellowship (PF-12-157-01-DMC). S.K. is an Investigator of the Howard Hughes Medical Institute. This work was supported by the Ministerio de Ciencia e Innovación (BFU2010-18965, BFU2013-43965-P and BFU2016-80370-P, I.R.), by the UAB-Aposta award to young investigators (APOSTA2011-03, I.R.) and by the NIH (R35 GM118175, to M.J. and R35 GM118092 to S.K.).

## Author contributions

S.P., M.M.-O., and I.R. conceived the experiments. O.F.-C. provided critical reagents. S. P., A.M.-L., M.M.-O, C.R., A.M.-M., J.F.-L., J.L., and I.R. performed the experiments. S. P., A.M.-L., M.M.-O., J.L., M.J., S.K., O.F.-C., M.G.-C., and I.R. analyzed the data. S.P., J. L., S.K., and I.R. wrote the manuscript.

## Additional information

**Competing interests:** The authors declare no competing interests.

