## [Peer Review File · Nature Communications]

Reviewers' comments:

Reviewer #1 (Remarks to the Author):

ATR is required to complete meiotic recombination in mice

Strengths/Highlights of study:

1. novel role for ATR in meiotic recombination
2. Genetic analysis validated with the pharmacological inhibition of ATR.
3. Phenotype severity can be modulated by altering amounts of ATR inhibitor.
4. Organ culture and pharmacological inhibition will potentially be a useful working model for the study of other drugable pathways/mechanisms that impact meiosis, especially given the inability to culture spermatocytes in culture over longer periods of time.

Comments/Critique:

1. Higher levels of the shorter Atr transcript ($\Delta E9$) can be seen accompanying depleted levels of full length Atr mRNA in the AtrS/S testes. While the authors have addressed the depletion of ATR levels by IF analysis on the sex body and by looking at the levels of $\gamma H2Ax$, it is not clear whether the shorter transcript is translated in spermatocytes and whether it might compensate for the loss of full length ATR. If it is possible to differentiate between the isoforms by SDS-PAGE and western blot, determining their relative levels in the mutant testes might significantly impact the conclusions.

2. The authors conclude that ATR is required for the assembly of RAD51 and DMC1 at RPA coated sites (DSB resected sites). They also show that ATR functions in synaptonemal complex (SC) extension/elongation. I therefore wonder whether changes in numbers of recombination foci seen upon the loss of ATR activity might be a direct consequence of defects in SC formation. It is not clear whether the reduction in foci translates to altered association with chromatin in general or to the mapped DSB site.

This can be done by:

- a. By determining the levels of DMC1 and RAD51 in the soluble vs insoluble chromatin by a simple nuclear fractionation of control and mutant animals (AtrS/S mice).
- b. RAD51 and DMC1 CHIP-PCR at a handful of mapped sites (Smagulova et al. 2011) in the control and mutant animals (AtrS/S mice).

4. It would be interesting to know the dynamics of pRPA (S33) levels, a known signal for MSCI/MSUC that is ATR dependent (Fedoriw et al. 2015).

Similarly, other markers of unpaired Chromatin such as pCHK1 (S317 & S345) can also be monitored in the ATR mutant males or chemically inhibited organ cultures.

5. As HORMAD1 plays a role in SC formation and is phosphorylated (pS375) presumably by ATR (Fukuda et al. 2012), it might be a good idea to look at pHORMAD1 distribution along autosomal asynapsed axes upon the inhibition of ATR activity.

6. The authors should do a better job of describing the method used to measure MLH1

distribution for experiment in fig.3d.

7. Does the decreased accumulation of RAD51 and DMC1 on asynapsed chromatin promote sister chromatid exchange? Can the authors perform sister chromatid exchange assays on cells isolated from *AtrS/S* males that are able to complete meiosis all be it with delayed kinetics. Alternatively it might be useful to perform the assay on organ culture treated with ATR inhibitors at lower doses.

References

Fedoriw AM, Menon D, Kim Y, Mu W, Magnuson T. 2015. Key mediators of somatic ATR signaling localize to unpaired chromosomes in spermatocytes. *Development*.

<http://dev.biologists.org/cgi/doi/10.1242/dev.126078>
<http://www.ncbi.nlm.nih.gov/pubmed/26209650>.

Fukuda T, Pratto F, Schimenti JC, Turner JM a, Camerini-Otero RD, Höög C. 2012. Phosphorylation of chromosome core components may serve as axis marks for the status of chromosomal events during mammalian meiosis. *PLoS Genet* 8.

Smagulova F, Gregoret I V, Brick K, Khil P, Camerini-Otero RD, Petukhova G V. 2011. Genome-wide analysis reveals novel molecular features of mouse recombination hotspots. *Nature* 472: 375–378.

Reviewer #2 (Remarks to the Author):

In this manuscript, Pacheco et al. analyze meiotic progression in *Seckel* mutant mice (*Atr* hypomorphs) alone and under various conditions of manipulation, in an attempt to shed light on meiotic functions of ATR, a major mammalian PIKK central to the DNA damage response.

This is a timely paper that complements other recent (yeast) studies that elucidate ATR function in meiosis. It is clearly structured and for the most part well-written. Analyses are thorough and of high quality. Unfortunately, however, these upsides do not overcome the fact that the experimental system is sub-optimal for the questions for thorough exploration of the most important questions regarding ATR's meiotic function. I wish I could be more positive about this manuscript as the authors really did their best to extract as much information as is possible with this mouse model.

The key caveat is that the low levels of ATR in these mice are sufficient to maintain relatively normal male meiosis. In hindsight, this is not a big surprise in light of Murga et al.'s earlier findings (*Nat Genet* 2009) of functional sperm in these animals.

Pacheco et al. go through substantial effort to lower ATR levels further, by both oral and *ex vivo* administration of ATRi. Both approaches had their problems, as the authors quite thoroughly discuss. Their main findings, while solid, are rather limited, demonstrating a role for ATR in RAD51/DMC1 focus formation, and indicating a function in crossover control.

Reviewer #3 (Remarks to the Author):

This manuscript describes the use of chemical inhibition and a Seckel patient-derived mutation to test the role of ATR in mouse spermatocytes. The authors show that the Seckel mutation leads to defects in the synapsis of sex chromosomes, changes in the number of repair foci, and a mild decrease in the number of MLH1-marked crossover sites. The reduction in repair foci in leptoneuma and zygonema is not the result of a decrease in DSB repair. Instead, experiments using chemical inhibitors of ATR and the downstream effector kinase CHK1 suggest an defect in the loading of the repair factor RAD51 onto RPA-coated break ends.

The paper is well written and the conclusions are supported by the presented data. ATR is a key regulator of chromosome integrity and the functional analysis of this protein during spermatogenesis will be of interest to a broad readership. A few points should be addressed, however, before this manuscript is considered for publication

1. Did the authors note any defects in sex chromosome synapsis in *AtrS/+* heterozygotes? The fact that the Seckel mutation expresses a truncated isoform of ATR raises the questions if this mutation confers dominant phenotypes.
2. The AZ20 treatment nicely recapitulates the Seckel observations for the reduction in RAD51 foci in leptoneuma and zygonema but the Seckel mutant has an increase in RAD52 counts in pachynema. Is the same true for pachytene spermatocytes in the AZ20 experiment or this effect dependent on the level of remaining ATR activity? Please include RAD51 counts for that stage for the AZ20 treatment.
3. Please remove the error bars for the top bars in Figure 1B and subsequent similar figures (Fig 4, 5). These error bars do not add data because they are by definition the same as the error bars of the lower bar. Also please indicate the genotype in the images for Fig 1B and C.
4. Figure 5: Why are the early leptoneuma numbers for RAD51 so different between the DMSO controls of F and J? Is this because of different DMSO concentrations?

Minor comments:

1. p5. Could the increased number of autosomal gammaH2AX patches not also indicate an increase in late forming breaks instead of delayed repair kinetics?
2. p7, top: Remove leftover comment
3. Fig 1D, 5E: the blue legend is not legible in the black background of the figures.
4. Supplementary Figure 1D: Please increase the signal for the sample images.

We appreciate the reviewers' constructive comments, which helped strengthen and clarify the paper. Our responses to their comments and suggestions are detailed below (reviewer comments in black text; responses in blue). We made substantial changes to the text and the Results and Discussion sections and we revised and added new figure panels (**Figure 3a, 4e, 4g and Supplementary Figure 1a, 1e and 1h**). The following major experimental changes were incorporated:

- We have been unable to detect the truncated ATR form predicted for the incorrectly spliced transcript in the Seckel mutant. Thus, we assume that, if translated, this isoform is rapidly degraded.
- We studied the synapsis of X and Y chromosomes in heterozygous *Atr*^{+/^S mice. The presence of one copy of the Seckel mutation had no observable effect on the ability of the X and Y chromosomes to find one another. These data suggest that the truncated ATR form is unlikely to be a dominant negative.}
- We analyzed the phosphorylation of three additional ATR targets that accumulate at the sex body at pachynema (S271 pHORMAD2, S33 pRPA, and S317 pCHK1). All three markers displayed changes in the staining intensity or spatial distribution, reinforcing our previous results showing that Seckel mutation impairs ATR function in spermatocytes.
- We expanded the analysis of late recombination markers by studying the formation of RNF212 foci in Seckel mice and AZ20-treated spermatocytes. In both models, ATR-deficient spermatocytes presented an increased number of RNF212 foci at pachynema. Furthermore, in both models, RNF212 foci tend to accumulate in clusters covering stretches of SC. These data strengthen our conclusion that wild type levels of ATR function are required for timely repair of double-strand breaks as crossovers.
- We expanded the analysis of RAD51 focus formation in AZ20-treated pachytene cells. This analysis again recapitulated our findings in Seckel mouse cells, as we observed an accumulation of RAD51 foci in AZ20-treated cells. These data support our conclusion that ATR function is required for meiotic recombination to occur in a timely manner.

Reviewers' comments:

Reviewer #1 (Remarks to the Author):

ATR is required to complete meiotic recombination in mice

Strengths/Highlights of study:

1. novel role for ATR in meiotic recombination
2. Genetic analysis validated with the pharmacological inhibition of ATR.
3. Phenotype severity can be modulated by altering amounts of ATR inhibitor.
4. Organ culture and pharmacological inhibition will potentially be a useful working model for the study of other drugable pathways/mechanisms that impact meiosis, especially given the inability to culture spermatocytes in culture over longer periods of time.

Comments/Critique:

1. Higher levels of the shorter *Atr* transcript ($\Delta E9$) can be seen accompanying depleted levels of full length *Atr* mRNA in the *Atr*^{S/S} testes. While the authors have addressed the depletion of ATR levels by IF analysis on the sex body and by looking at the levels of γ H2Ax, it is not clear whether the shorter transcript is translated in spermatocytes and whether it might

compensate for the loss of full length ATR. If it is possible to differentiate between the isoforms by SDS-PAGE and western blot, determining their relative levels in the mutant testes might significantly impact the conclusions.

Although we understand the concern of reviewer 1, we think it is unlikely that the truncated ATR form may compensate for the loss of full-length ATR. This is because the kinase catalytic domain of ATR (encoded in exons 39 to 47) is located at the C-terminus of the protein and the Seckel mutation causes the introduction of a stop codon before exon 10. Nonetheless, we acknowledge this fact was not well addressed in our manuscript, so we added more details in the introduction of the Seckel mutation (page 4). Furthermore, we analyzed by western blot if Seckel mouse testes present a truncated protein of the expected weight (~70 kDa). To do so, we used an antibody against an ATR peptide located at the N-terminus of the protein (see Materials and Methods for more details). In all cases, we have been unable to see a specific band in the Seckel mutant that was not present in the wild type samples (see the new **Supplementary Figure 1a** for an example). Thus, we conclude that if the $\Delta E9$ transcript is translated, the polypeptide is degraded rapidly. Furthermore, this western blot analysis shows something that had already been reported by Murga et al (2009), which is that global ATR protein levels in adult Seckel mouse testis are comparable to wild type (**Supplementary Figure 1a**). Nonetheless, it is important to note that the amount of protein present in the sex chromosome chromatin of Seckel mouse spermatocytes is significantly reduced compared to control spermatocytes (**Fig. 1b**), thus implying that global levels of ATR protein from whole testis extracts do not correlate well with the amount of protein specifically in spermatocytes.

2. The authors conclude that ATR is required for the assembly of RAD51 and DMC1 at RPA coated sites (DSB resected sites). They also show that ATR functions in synaptonemal complex (SC) extension/elongation. I therefore wonder whether changes in numbers of recombination foci seen upon the loss of ATR activity might be a direct consequence of defects in SC formation.

It is true that meiotic recombination and homologous chromosome synapsis are tightly linked processes. However, our data clearly indicate that the reduction of RAD51 and DMC1 foci cannot be a consequence of the observed SC defects seen. First, a reduction in RAD51 and DMC1 foci is already apparent at early leptotema, which is before any SC formation has been initiated. Second, our analysis with ATR inhibitors in *Spo11* mutant mice reveals that ATR has no role in SC initiation per se. Finally, we note that prior studies have established that defects in SC formation cause an increase in numbers of recombination foci, not a decrease, because SC formation is normally accompanied by cessation of DSB formation (Kauppi et al., *Genes Dev* 2013).

It is not clear whether the reduction in foci translates to altered association with chromatin in general or to the mapped DSB site.

This can be done by:

- a. By determining the levels of DMC1 and RAD51 in the soluble vs insoluble chromatin by a simple nuclear fractionation of control and mutant animals (*AtrS/S* mice).
- b. RAD51 and DMC1 CHIP-PCR at a handful of mapped sites (Smagulova et al. 2011) in the control and mutant animals (*AtrS/S* mice).

It is not clear to us what scenario the reviewer has in mind with this suggestion. It is well established in the field that foci of RAD51 and DMC1 assemble specifically at sites of SPO11-dependent DSBs during meiosis. Prior studies have shown that focus formation is almost completely eliminated in the absence of SPO11-mediated DSBs (with the very small number of residual foci suggested to be the result of transposon-mediated DNA damage). Thus, it is well

accepted that focal accumulation of RAD51 and DMC1 is at DSBs, and we are not aware of any studies implicating general association with chromatin, i.e., not at DSBs, as a meaningful contributor to cytologically observable complexes of these proteins in mice. Our data clearly demonstrate that fewer foci are present early in prophase, whereas SPO11-oligo complexes are not reduced and RPA foci are elevated. These data are thus strong prima facie evidence that SPO11-dependent DSBs form relatively normally but assembly of DMC1 and RAD51 at these sites is impaired.

Importantly, we emphasize that all available data from many researchers clearly tie RAD51 and DMC1 foci to sites of DSBs within individual cells, not simply to “mapped DSB sites” in the genome. Mapped DSB sites (known as hotspots) are a population-average measure of the targeting of the DSB machinery by PRDM9. It is immaterial for our purposes whether the DSBs in the Seckel mutant map to the same genomic coordinates as in wild type, since the defect we are examining pertains to what happens at individual DSBs once they have formed. Moreover, the Seckel mutant does not resemble mutants lacking PRDM9, so there is no reason at present to consider altered DSB site choice as a candidate to explain our findings. The ChIP experiments proposed would be a substantial undertaking but would not, in our opinion, contribute to the understanding of the roles of ATR in promoting RAD51/DMC1 focus formation, so we have elected not to perform these experiments.

Nevertheless, to try to address the reviewer’s question, we analyzed RAD51 in testis extracts from wild type and Seckel mutant mice (see figure below). Both genotypes yielded a similar proportion of RAD51 in both chromatin-rich and chromatin-poor fractions. Thus, there is no evidence for a reduced ability for RAD51 to associate with chromatin in Seckel mice. We also emphasize that we observed wild type numbers of RAD51 foci at late zygonema in mutant cells and an accumulation of RAD51 foci at pachynema. The data thus clearly indicate that Seckel mutant cells are able to load RAD51 into resected DSBs, but it takes them longer to do so.

Figure 1: RAD51 is present on both chromatin-rich and -poor fractions in control and Seckel samples. WTE: Whole testis extracts. Chr+: Chromatin-rich fraction. Chr-: Chromatin-poor fraction.

4. It would be interesting to know the dynamics of pRPA (S33) levels, a known signal for MSCI/MSUC that is ATR dependent (Fedoriw et al. 2015). Similarly, other markers of unpaired Chromatin such as pCHK1 (S317 & S345) can also be monitored in the ATR mutant males or chemically inhibited organ cultures.

Thank you for this suggestion. To address this, we analyzed the presence of S33 pRPA and S317 pCHK1 in the sex bodies of *Atr* mutant spermatocytes (see new **Supplementary Figure 1f**). Whereas wild-type spermatocytes present both markers all over the chromatin of the X and Y chromosomes, cells from Seckel mice show staining that is mostly confined around the X and Y chromosome axes. These staining patterns resemble the ATR staining in Seckel mutant

spermatocytes (**Fig. 1b**). Thus, these data reinforce our previous conclusion that wild-type levels of ATR protein are required to maintain the characteristic marks of the sex chromosome chromatin.

5. As HORMAD1 plays a role in SC formation and is phosphorylated (pS375) presumably by ATR (Fukuda et al. 2012), it might be a good idea to look at pHORMAD1 distribution along autosomal asynapsed axes upon the inhibition of ATR activity.

Unfortunately, we could not analyze the dynamics of HORMAD1 S375 phosphorylation due to antibody unavailability. However, we were able to obtain an antibody that specifically recognizes the phosphorylated form of HORMAD2 (S271), which is also ATR-dependent (Royo et al 2013). The HORMAD2 phosphorylation dynamics was indistinguishable between Seckel mutant and wild-type spermatocytes, thus it is not obvious if this particular phosphorylation plays a role in linking ATR function to SC elongation. We also quantified the intensity of the pHORMAD2 signal on the sex chromosome axes at pachynema. Seckel mutants presented a slightly higher intensity of pHORMAD2 than wild type (**Supplementary Figure 1e**). We think this may be a result of the presence of ATR mostly on the X and Y chromosome axes in Seckel mouse spermatocytes.

6. The authors should do a better job of describing the method used to measure MLH1 distribution for experiment in fig.3d.

We apologize for the brief description that was in our previous draft. We included a new section in the Materials and Methods (page 20), in which we describe how we analyzed the data in prior Fig. 3d, now **Figs. 3e and 3f**.

7. Does the decreased accumulation of RAD51 and DMC1 on asynapsed chromatin promote sister chromatid exchange? Can the authors perform sister chromatid exchange assays on cells isolated from *AtrS/S* males that are able to complete meiosis all be it with delayed kinetics. Alternatively it might be useful to perform the assay on organ culture treated with ATR inhibitors at lower doses.

We do not think that decreased RAD51 and DMC1 would be likely to cause increased sister chromatid exchange (SCE) because these recombinases are presumably also required for SCE, as in yeast. However, we agree that the reverse direction of causality is plausible in principle: if ATR is involved in meiotic recombination partner choice by favoring the homologous chromosome, then increased use of the sister chromatid instead could cause decreased levels of RAD51 and DMC1 foci because of more rapid turnover of recombination intermediates (assuming sister chromatid recombination is faster than interhomolog). We do not favor this interpretation because it does not predict the increase in RPA foci that we observe. Our preferred interpretation of a delay in loading of RAD51 and DMC1 provides a more parsimonious explanation for all of the findings.

Unfortunately, the proposed experiment does not seem realistic for mouse testis with currently available technology. As far as we know, no one has ever performed SCE assays on mouse meiocytes, and several issues make this experiment technically challenging. First, SCE assays require growing cells in the presence of BrdU for two cell divisions, obtaining metaphase-stage cells, and analyzing the occurrence of exchange between the two sister chromatids of each chromosome at metaphase. Thus, it is common to use synchronized cultures to do this analysis. In order to perform SCE in mouse spermatocytes one would need synchronous meiotic cells/cultures. Even though the first juvenile wave of meiosis is considered to be synchronized, it is not in fact synchronous enough to perform SCE assays.

Second, the time required to expose spermatogonia to BrdU during two rounds of DNA replication before initiation of meiosis is not known, either *in vivo* or *in vitro*. Third, the fact that sister chromatids are maintained together until anaphase I would most likely preclude the use of metaphase I spermatocytes for this purpose. Moreover, the rapid transition through the second meiotic division could result in a very low amount of metaphase II spermatocytes that could certainly make the analysis very difficult. Because of these technical limitations, we decided not to pursue this experiment.

Reviewer #2 (Remarks to the Author):

In this manuscript, Pacheco et al. analyze meiotic progression in Seckel mutant mice (*Atr* hypomorphs) alone and under various conditions of manipulation, in an attempt to shed light on meiotic functions of ATR, a major mammalian PIKK central to the DNA damage response.

This is a timely paper that complements other recent (yeast) studies that elucidate ATR function in meiosis. It is clearly structured and for the most part well-written. Analyses are thorough and of high quality. Unfortunately, however, these upsides do not overcome the fact that the experimental system is sub-optimal for the questions for thorough exploration of the most important questions regarding ATR's meiotic function. I wish I could be more positive about this manuscript as the authors really did their best to extract as much information as is possible with this mouse model.

The key caveat is that the low levels of ATR in these mice are sufficient to maintain relatively normal male meiosis. In hindsight, this is not a big surprise in light of Murga et al.'s earlier findings (Nat Genet 2009) of functional sperm in these animals.

Pacheco et al. go through substantial effort to lower ATR levels further, by both oral and ex vivo administration of ATRi. Both approaches had their problems, as the authors quite thoroughly discuss. Their main findings, while solid, are rather limited, demonstrating a role for ATR in RAD51/DMC1 focus formation, and indicating a function in crossover control.

We thank the reviewer for the appreciation of our work and for considering our findings as solid. However, we disagree that our genetic and pharmacological models are sub-optimal for exploring the most important questions regarding ATR's meiotic functions. The main caveat reviewer 2 found in our study is that ATR levels in the Seckel mutant are not low enough to block normal male meiosis and thus this invalidates the use of this mutant for the analysis. We certainly agree that, as a general rule, the best mutant mice to study the function of a protein in mammalian meiosis are those that completely abolish the expression of the gene of interest. Unfortunately, that is not always an option since many genes involved in meiosis are also required for embryonic development. In such circumstances, hypomorphic mutations that reduce the expression of the gene of interest to a point where a phenotype can be observed become a particularly important resource. There is a long history of genetic inquiry using hypomorphs to query the functions of essential genes. We would like to emphasize a key point in this context: if a hypomorphic mutation does not affect a biological process, one cannot conclude that the gene is dispensable. But if the mutation does cause an altered phenotype, as in our study, there is no ambiguity as to whether the gene is involved. In our particular case, even though Seckel mouse mutants are able to complete spermatogenesis, they present subtle meiotic defects. More importantly, these defects can also be observed when mice are treated with ATR inhibitors, which validates their correlation to ATR function.

Furthermore, all our findings have been corroborated by a study co-submitted with ours by Dr. James Turner, in which a conditional mutation was used to analyze ATR meiotic function (Widger et al, under review; available on BiorXiv). As predicted from previous studies, and because of the severe reduction of *Atr* expression, this conditional mutation causes a meiotic arrest. Mutant spermatocytes are unable to form a sex body and this causes apoptosis. Therefore, the most advanced spermatocytes found are mid-pachytene cells.

The activation of this meiotic arrest is relevant because it prevents the use of this mutant to study of the role of ATR in later meiotic stages, for instance, in mammalian crossover formation and distribution. Because yeast and *Drosophila* ATR homologs (*Mec1* and *Mei-41*

respectively) have roles in crossover formation and distribution, we consider this an important topic to address. Our use of the hypomorphic mutation plus the pharmacological approach has allowed us to study MLH1 foci formation and distribution in ATR-deficient spermatocytes. Moreover, we expanded our study to another late recombination marker, RNF212, which is also required to form crossovers, to gain more details about this process. Our new data (Fig. **3b** and **4g**), obtained in Seckel mice and confirmed by the pharmacological inhibition of ATR *in vivo*, show that ATR function is required for the timely reduction in RNF212 foci from synapsed chromosomes at pachynema. These data, along with:

- 1) The accumulation of RAD51 and DMC1 at ATR-deficient pachynema
- 2) The reduced formation of MLH1 foci in Seckel and AZ20-treated cells
- 3) The normal distribution of MLH1 foci along the bivalents
- 4) The normal MLH1 interfocal distances in ATR-deficient spermatocytes
- 5) The normal proportion of achiasmate chromosomes at diplonema in Seckel mouse spermatocytes

make us suggest that full levels of ATR function are not required to form crossovers or to properly distribute them along the genome (on a cytological scale). Nonetheless, our data shows that wild-type levels of ATR function are required for timely formation of COs at late pachynema. It is worth noting that these results could not be inferred from any other data obtained in the conditional mutant, and thus emphasizes the utility of our hypomorphic and pharmacological approach to study the ATR functions during later stages of meiotic prophase.

Finally, our study also addresses two other points, which in our opinion are relevant to the broad audience of *Nature Communications*. First, understanding in detail the phenotypic consequences of the Seckel mutation is of direct relevance to those human populations in which the mutation is present at appreciable frequency. Second, our studies reveal the effect that oral administration of ATR inhibitors may have on male gametogenesis. The clear reduction in the presence of diplotene spermatocytes in treated animals suggest that the sustained administration of ATRi to male mice may halt spermatogenesis and thus cause at least transient infertility. This finding is relevant to predicting potential side effects that the use of ATRi may have if these drugs are used in humans.

Reviewer #3 (Remarks to the Author):

This manuscript describes the use of chemical inhibition and a Seckel patient-derived mutation to test the role of ATR in mouse spermatocytes. The authors show that the Seckel mutation leads to defects in the synapsis of sex chromosomes, changes in the number of repair foci, and a mild decrease in the number of MLH1-marked crossover sites. The reduction in repair foci in leptoneuma and zygonema is not the result of a decrease in DSB repair. Instead, experiments using chemical inhibitors of ATR and the downstream effector kinase CHK1 suggest an defect in the loading of the repair factor RAD51 onto RPA-coated break ends.

The paper is well written and the conclusions are supported by the presented data. ATR is a key regulator of chromosome integrity and the functional analysis of this protein during spermatogenesis will be of interest to a broad readership. A few points should be addressed, however, before this manuscript is considered for publication

1. Did the authors note any defects in sex chromosome synapsis in *AtrS*/+heterozygotes? The fact that the Seckel mutation expresses a truncated isoform of ATR raises the questions if this mutation confers dominant phenotypes.

Thank you for this suggestion. Upon further analysis, we found a comparable number of spermatocytes with unsynapsed X and Y in wild type and heterozygous mice (manuscript page 5). Thus, there is no evidence to suggest that the truncated ATR form has a dominant negative effect. Furthermore, we were unable to detect the truncated form in whole testis protein extracts from Seckel mice (new **Supplementary Figure 1a**). We feel that this experiment strengthens the conclusions from our study.

2. The AZ20 treatment nicely recapitulates the Seckel observations for the reduction in RAD51 foci in leptoneuma and zygonema but the Seckel mutant has an increase in RAD52 counts in pachynema. Is the same true for pachytene spermatocytes in the AZ20 experiment or this effect dependent on the level of remaining ATR activity? Please include RAD51 counts for that stage for the AZ20 treatment.

We performed the suggested analyses and included them in the manuscript (**Fig. 4e** and **Supplementary Table 2**). Indeed, AZ20 treatment again recapitulated the accumulation of RAD51 foci observed in the Seckel mice pachytene cells, reinforcing our previous conclusions.

3. Please remove the error bars for the top bars in Figure 1B and subsequent similar figures (Fig 4, 5). These error bars do not add data because they are by definition the same as the error bars of the lower bar. Also please indicate the genotype in the images for Fig 1B and C.

We made the suggested change in Fig. **1B**, **4** and **5** and added the genotypes for Fig. **1B** and **C**.

4. Figure 5: Why are the early leptoneuma numbers for RAD51 so different between the DMSO controls of F and J? Is this because of different DMSO concentrations?

As the reviewer guessed, the two DMSO concentrations are different (0.1% and 2.1%). We think this difference could slightly alter focus numbers. However, we note that all cells in the AZ20 analysis (Fig. **5f**) are within the range found in the CHK1i analysis (Fig. **5j**), and the difference between the DMSO controls in the two experiments is not statistically significant ($p > 0.005$, t test).

Minor comments:

1. p5. Could the increased number of autosomal gammaH2AX patches not also indicate an increase in late forming breaks instead of delayed repair kinetics?

We agree that this is a formal possibility, so we included this option in the text (page 5).

2. p7, top: Remove leftover comment

Thank you; done.

3. Fig 1D, 5E: the blue legend is not legible in the black background of the figures.

We modified the legend to make it more readable.

4. Supplementary Figure 1D: Please increase the signal for the sample images.

We modified the figure (now Supplementary Fig. 1e) as suggested.

REVIEWERS' COMMENTS:

Reviewer #1 (Remarks to the Author):

As stated in the original review the strengths of this manuscript are

1. novel role for ATR in meiotic recombination
2. Genetic analysis validated with the pharmacological inhibition of ATR.
3. Phenotype severity can be modulated by altering amounts of ATR inhibitor.
4. Organ culture and pharmacological inhibition will potentially be a useful working model for the study of other drugable pathways/mechanisms that impact meiosis, especially given the inability to culture spermatocytes in culture over longer periods of time.

The authors have presented very thoughtful responses and have conducted additional experiments, when possible, to substantiate their results. Their responses and additional data/changes have strengthened the manuscript. The results will be of interest to a broad community.

Reviewer #3 (Remarks to the Author):

The authors have adequately addressed my concerns. The inclusion of RNF212 is a nice addition to this paper that further strengthens the conclusions. This paper is expected to be of interest to the field and the presented findings have important clinical implications.

REVIEWERS' COMMENTS:

Reviewer #1 (Remarks to the Author):

As stated in the original review the strengths of this manuscript are

1. novel role for ATR in meiotic recombination
2. Genetic analysis validated with the pharmacological inhibition of ATR.
3. Phenotype severity can be modulated by altering amounts of ATR inhibitor.
4. Organ culture and pharmacological inhibition will potentially be a useful working model for the study of other drugable pathways/mechanisms that impact meiosis, especially given the inability to culture spermatocytes in culture over longer periods of time.

The authors have presented very thoughtful responses and have conducted additional experiments, when possible, to substantiate their results. Their responses and additional data/changes have strengthened the manuscript. The results will be of interest to a broad community.

Reviewer #3 (Remarks to the Author):

The authors have adequately addressed my concerns. The inclusion of RNF212 is a nice addition to this paper that further strengthens the conclusions. This paper is expected to be of interest to the field and the presented findings have important clinical implications.

We thank reviewers #1 and #3 for their appreciation of our work.